# Pandemic H1N1 influenza A viruses suppress immunogenic RIPK3-driven dendritic cell death

Boris M. Hartmann [1], Randy A. Albrecht [2], Elena Zaslavsky[1], German Nudelman[1], Hanna Pincas[1], Nada Marjanovic[1], Michael Schotsaert[2], Carles Martínez-Romero [2], Rafael Fenutria[2], Justin P. Ingram[3], Irene Ramos[2], Ana Fernandez-Sesma[2], Siddharth Balachandran[3], Adolfo García-Sastre[2,4] & Stuart C. Sealfon[1]

The risk of emerging pandemic influenza A viruses (IAVs) that approach the devastating 1918 strain motivates finding strain-specific host–pathogen mechanisms. During infection, dendritic cells (DC) mature into antigen-presenting cells that activate T cells, linking innate to adaptive immunity. DC infection with seasonal IAVs, but not with the 1918 and 2009 pandemic strains, induces global RNA degradation. Here, we show that DC infection with seasonal IAV causes immunogenic RIPK3-mediated cell death. Pandemic IAV suppresses this immunogenic DC cell death. Only DC infected with seasonal IAV, but not with pandemic IAV, enhance maturation of uninfected DC and T cell proliferation. In vivo, circulating T cell levels are reduced after pandemic, but not seasonal, IAV infection. Using recombinant viruses, we identify the HA genomic segment as the mediator of cell death inhibition. These results show how pandemic influenza viruses subvert the immune response.

[1] Department of Neurology, Icahn School of Medicine at Mount Sinai, New York, NY 10029, USA. [2] Department of Microbiology and Global Health & Emerging Pathogens Institute, Icahn School of Medicine at Mount Sinai, New York, NY 10029, USA. [3] Fox Chase Cancer Center, Philadelphia, PA 19111, USA. [4] Department of Medicine, Division of Infectious Diseases, Icahn School of Medicine at Mount Sinai, New York, NY 10029, USA. Correspondence and requests for materials should be addressed to S.C.S. (email: stuart.sealfon@mssm.edu)

nfluenza A virus (IAV) is a major pathogen for humans and other species. Each year, seasonal IAVs cause epidemics that affect 5–15% of the population with upper respiratory tract infections. Although this results in hundreds of thousands of deaths globally, mainly among high-risk groups (very young, elderly, and chronically ill), most cases are mild. In contrast, pandemic IAVs such as the 1918 strain that killed approximately 50 million people worldwide[1], can be associated with much higher rates of infection and mortality. The severity of disease outcome is influenced by the virulence of the IAV strain[2, 3], an important component of which comprises the mechanisms used by each strain to interfere with host defenses[4–6].

Following infection of lung epithelial cells, IAV spreads to both nonimmune and innate resident respiratory tract immune cells, including dendritic cells (DC) (for review, see ref. [7]). DC play a pivotal role in the initiation of IAV immunity (for review, see ref. [8]).

They detect specific components of viral particles termed pathogen-associated molecular patterns (PAMPs) through their pattern-recognition receptors, which include Toll-like receptors, RIG-I-like receptors, and (NOD)-like receptors[9]. PAMP recognition induces intracellular signaling cascades that lead to the secretion of type I interferons (IFNs) and proinflammatory cytokines. Proinflammatory cytokines, other danger signals, and/or direct contact with infected DC initiate the maturation of uninfected DC[10] into professional antigen-presenting cells that pick up and process viral antigen, and migrate to the lymph nodes, where activation of T cells helps mediate adaptive immunity (for review, see ref. [11]). DC maturation involves morphological changes, loss of endocytic/phagocytic receptors, secretion of cytokines and chemokines, as well as upregulation of various adhesion, homing (e.g., CCR7), co-stimulatory (e.g., CD80 and CD86), and MHC class I and II surface molecules that mediate antigen presentation[11].

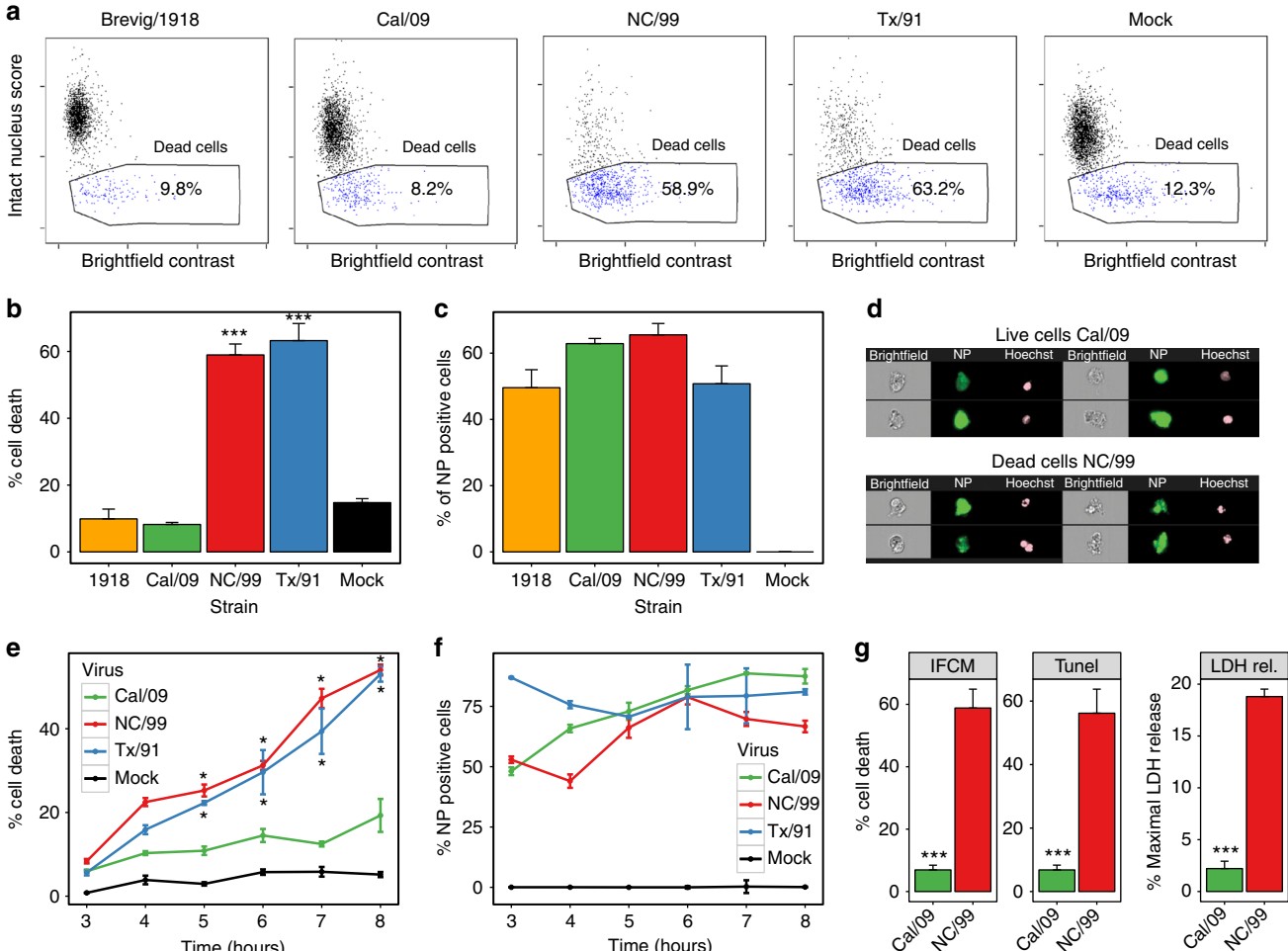

**Fig. 1** Seasonal but not pandemic IAVs induce human DC death. DC were infected for 8 h with either mock or the indicated IAV strains. Cell death was assayed through quantification of nuclear fragmentation and assessment of cell morphology by imaging flow cytometry. The Brevig/1918 and Cal/09 are pandemic IAV strains, whereas the NC/99 and Tx/91 are seasonal IAV strains. **a** Flow cytometry plots illustrating the percentage of dead cells after DC infection. Percentage of cell death (**b**) and of NP-positive cells (**c**) following infection. ***$p < 0.001$. **d** Sample images of live cells following Cal/09 infection, and dead cells following NC/99 infection. Data are representative of three independent experiments using DC derived from different donors. $n = 3$ for experimental groups and for mock-infected controls. Time-course measurements of the percentages of cell death (**e**) and of the proportion of NP-expressing cells following DC infection (for 3–8 h) (**f**). *$p < 0.005$. Data are representative of four independent experiments using DC derived from different donors. $n = 3$ for experimental groups and for mock-infected controls. **g** Comparison of the percentage of cell death measurements obtained by imaging flow cytometry (IFCM), TUNEL method, and lactate dehydrogenase release cytotoxicity assay (LDH rel.). Note that the percentage of cell death in untreated cells and in infected cells varies in DC obtained from different individual donors. Data shown in **a–g** were obtained in DC from three different donors, respectively. ***$p < 0.005$. This experiment was performed three times. $n = 3$ technical replicates. **b**, **c**, **e–g** Values shown are median ± s.e.m, ANOVA followed by Tukey's honest significant difference (HSD) test

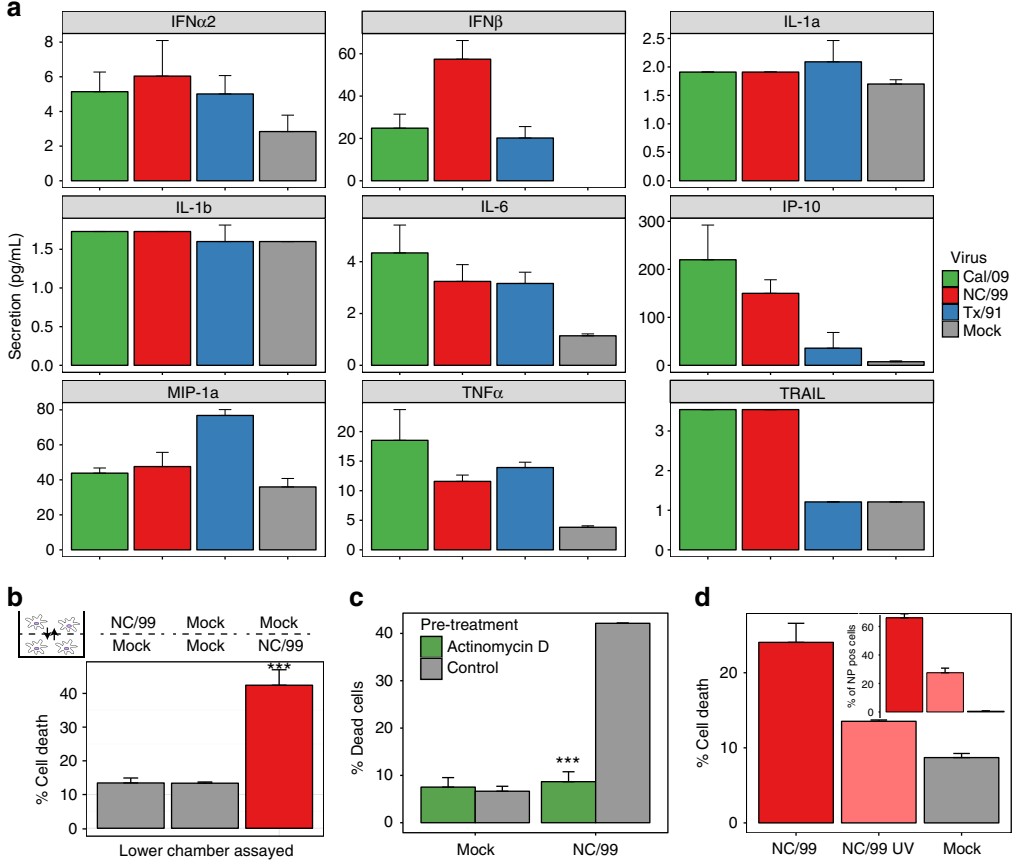

**Fig. 2** Cell death induced by seasonal NC/99 IAV depends on viral RNA replication. **a** Cytokine and chemokine release 4 h after DC infection with either mock, seasonal (NC/99; Tx/91), or pandemic (Cal/09) strains. Supernatants were collected, and cytokine and chemokine levels were measured by multiplex ELISA. The levels of IL-1a, IL-1b, TRAIL, and IL-6 are near or below the limits of detection for this assay. **b** Transwell experiment (see inset diagram) where mock-infected DC (lower chamber) were exposed to paracrine signaling from NC/99- or mock-infected DC (upper chamber). NC/99-infected cells in the lower chamber were used as a positive control. **c** Effect of a 30-min pretreatment with actinomycin D (2 µM) on virus-induced cell death. **d** Effect of reduced infectivity via UV irradiation (NC/99 UV) on virus-induced cell death. The inset shows the percentage of NP-positive cells under the different conditions. **b**, **c**, **d** DC were infected for 8 h with either mock or NC/99 IAV. Percentage of cell death was determined by imaging flow cytometry. Data are representative of two **a**, **d** or three **b**, **c** independent experiments using DC derived from different donors. $n = 3$ for experimental groups and for mock-infected controls. ***$p < 0.001$, ANOVA followed by Tukey's HSD test. Values shown are median ± s.e.m

Differences in IAV tropism and pathogenicity influence the capacity of the virus to induce host cell death (for review, see ref. [12]). While the role of programmed cell death via caspase-dependent apoptosis has been most extensively examined[13–16], programmed necrosis (or necroptosis) has recently emerged as a more immunogenic host cell death mechanism (for review, see ref. [17]). Unlike apoptosis, necroptosis results in the release of danger-associated molecular patterns (DAMPs), and thus is associated with inflammation and immune cell activation. Necroptosis is receptor-interacting protein kinase (RIPK)- and mixed lineage kinase domain-like pseudokinase (MLKL)-dependent. Depending on the host/pathogen context, necroptosis can either be involved in host response to infection or be exploited by the pathogen for further dissemination. cIAP2 (cellular inhibitor of apoptosis proteins-2) may inhibit necroptosis of pulmonary epithelial cells in mice infected with H1N1/PR8, thereby promoting host recovery[18]. On the other hand, our recent results demonstrate that RIPK3-induced apoptotic and necroptotic pathways are both activated by IAV, and important for the control of IAV spread in mice[19].

Studies in rodents have provided suggestions about differences in virus–host interactions that could contribute to differences in pathogenicity between seasonal and pandemic strains. Viruses containing the hemagglutinin (HA) and neuraminidase (NA) of

the 1918 human IAV are highly pathogenic in mice, resulting in high morbidity and mortality[20]. Infection with a mouse-adapted strain of the 2009 H1N1 pandemic virus also causes a higher mortality rate and cytokine response than infection with the seasonal-related mouse-adapted PR8 IAV strain[21]. However, comparative strain studies in mice may not accurately reflect the virus–host interaction differences of human strains, and mouse-adaptation may further distort the fidelity of this model for representing human IAV strain-specific mechanisms contributing to the effects of infection in humans. For example, cultured mouse DC, in contrast with human DC, are nearly resistant to infection by nonadapted human wild-type IAV strains[22]. Therefore, it is critical to identify and study human IAV strain-specific mechanisms for virus–host interactions using human models.

We previously reported that infection of human DC with seasonal H1N1 IAV strains (NC/99, Tx/91) caused rapid host RNA degradation, while infection with two related pandemic strains (1918, Cal/09) did not[23]. In the present work, we investigate the mechanisms underlying these differences. We demonstrate that this global RNA loss is linked to induction of RIPK3-mediated cell death by seasonal IAV strains. Conversely, pandemic IAV strains inhibit DC death. While DC death increases the proliferation of allogeneic T cells, pandemic IAV infection correlates with reduced T cell proliferation in vitro and lower T cell levels in human

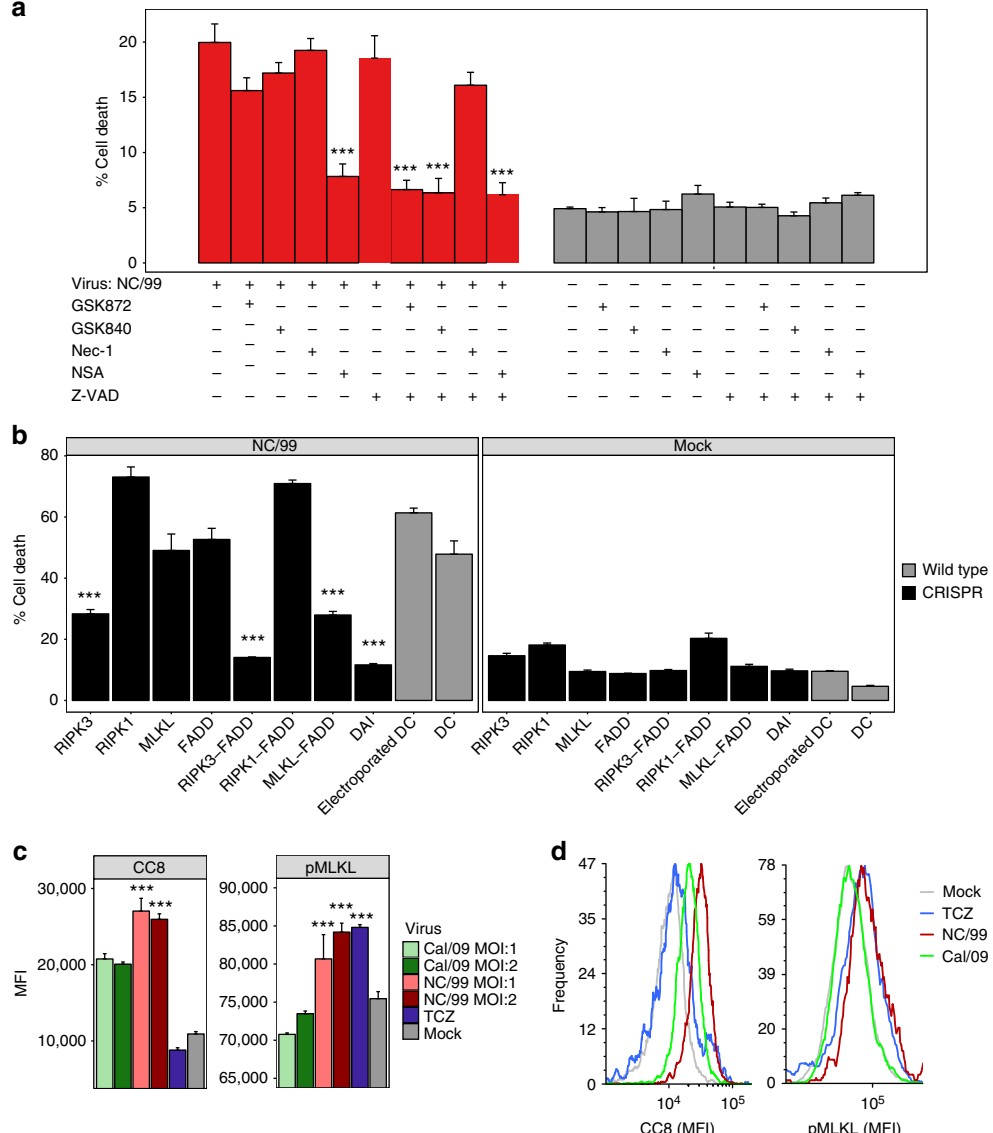

**Fig. 3** Seasonal NC/99 IAV induces RIPK3-dependent apoptosis and necroptosis. **a** DC infected for 8 h with either mock or NC/99 IAV were pretreated for 1 h with various pharmacological inhibitors: Z-VAD-FMK (20 μM), necrosulfonamide (NSA; 15 μM), GSK872 (5 μM), GSK840 (5 μM), and necrostatin (Nec-1; 30 μM). Percentage of cell death was determined by imaging flow cytometry. **b** Genetic ablation of RIPK1, RIPK3, MLKL, FADD, and DAI, either alone or in combination, was carried out by electroporation of DC with CRISPR/Cas9 ribonucleoproteins. Electroporated DC (negative control), DC electroporated with Cas9 Nuclease in the absence of targeting guide RNAs. Following an 8-h infection with either mock or NC/99 IAV, cell death was assayed in DC selected for having low levels of expression of the ablated gene(s) by imaging flow cytometry analysis (Supplementary Fig. 3). **c, d** Imaging flow cytometry measurements of MLKL phosphorylation and caspase 8 cleavage following a 5-h infection with either mock, NC/99, or Cal/09 IAV at the indicated multiplicities of infection (MOIs). DC treated for 5 h with the combination of TNFα (0.1 μg/ml), cycloheximide (2 μg/ml), and Z-VAD-FMK (50 μM; TCZ), a necroptosis inducer, were used as positive control. MFI, mean fluorescence intensity. **c** is a summary plot; **d** shows representative imaging flow cytometry plots. **a** Data are representative of two independent experiments with all inhibitors using DC derived from different donors. Additional experiments employing combinations of Z-VAD-FMK and one or two of the other inhibitors were repeated 3–8 times using cells from different donors. $n = 3$ for experimental groups and for mock-infected controls. **b, c** Data are representative of two independent experiments using cells from different donors. $n = 3$ for experimental groups and for controls. ***$p < 0.001$, ANOVA followed by Tukey's HSD test, with an additional Bonferroni multiple-testing correction for summarized data. Values shown are median ± s.e.m

infection in vivo. Importantly, we identify the pandemic HA viral segment as the sequence-specific determinant of cell death inhibition. Our findings reveal a novel mechanism, namely HA-mediated immunogenic cell death inhibition, by which pandemic IAV strains may evade the host immune response.

## Results

**Seasonal but not pandemic IAVs induce human DC death.** We sought to determine whether the previously reported divergent

effects of pandemic vs. seasonal strains on host RNA degradation[23] were linked to alterations in DC survival. We estimated the percentage of cell death based on the number of cells showing nuclear fragmentation and an altered cell morphology by imaging flow cytometry analysis (Fig. 1a, d) of primary human monocyte-derived DC (Supplementary Fig. 1). Paralleling the strain-specific induction of RNA degradation[23], infection by each seasonal strain (NC/99, Tx/91), but by neither pandemic strain (1918, Cal/09), induced cell death (Fig. 1a, b, d). Cell death was induced by the

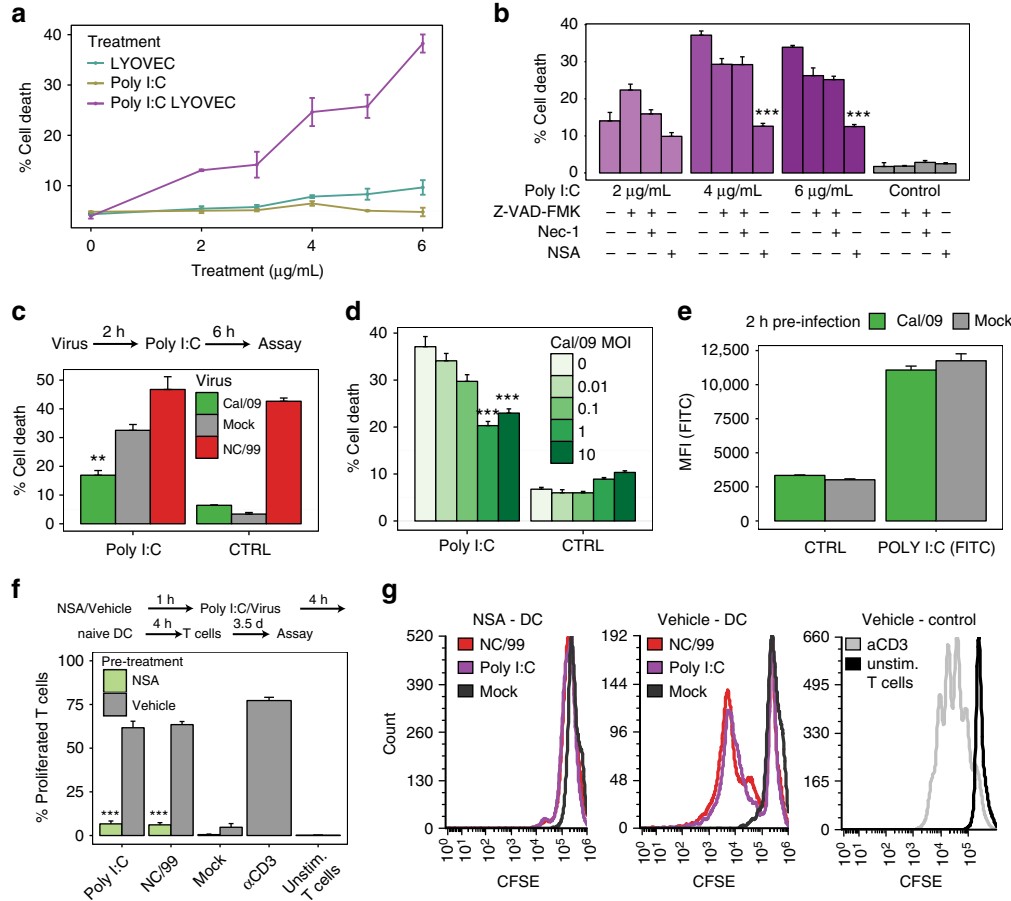

**Fig. 4** Pandemic Cal/09 IAV impairs poly I:C-induced cell death. **a** DC were treated for 8 h with increasing concentrations of either poly I:C, poly I:C combined with the transfecting agent LYOVEC, or LYOVEC alone. **b** DC were pretreated with the indicated pharmacological inhibitors for 1 h, and then transfected for 6 h with various concentrations of poly I:C (with LYOVEC) or with control. **c** DC were infected for 2 h with either mock, NC/99, or Cal/09 IAV, and then transfected with either poly I:C (4 μg/ml) or control for 6 h. **d** DC were infected for 2 h with Cal/09 at increasing MOIs, and then transfected with either poly I:C (4 μg/ml) or control for 6 h. **a–d** Percentage of cell death was determined by imaging flow cytometry. **e** DC were infected with either mock or Cal/09 for 2 h, and then transfected with either GFP-labeled poly I:C (4 μg/ml) or vehicle for 2 h. Transfection efficiency was determined by imaging flow cytometry via measurement of GFP expression. FITC, fluorescein isothiocyanate. **f, g** DC were pretreated for 1 h with either 15 μM NSA or vehicle, and either transfected with poly I:C (4 μg/ml) or infected with NC/99 or mock for 4 h. DC were then washed, cocultured with naive autologous DC for 4 h, and cocultured with carboxyfluorescein succinimidyl ester (CFSE)-labeled allogeneic T cells for 3.5 days. The percentage of proliferated T cells was determined by imaging flow cytometry as a decrease in cell fluorescence. T cells incubated with anti-CD3/CD28-coated beads served as a positive control. **f** is a summary plot; **g** shows representative imaging flow cytometry plots. Data shown in different panels were obtained using cells from different donors. Note that the absolute percentage of cell death induced by poly I:C transfection or virus infection can vary across experiments. Data are representative of three **a, b, d–f** or five **c** independent experiments using cells from different donors. n = 3 for experimental groups and for controls. **a, b, d–f** ***p < 0.001; **c** **p < 0.005, ANOVA followed by Tukey's HSD test, with an additional Bonferroni multiple-testing correction for summarized data. Values shown are median ± s.e.m

seasonal strains NC/99 and Tx/91 as early as 4 h after infection (Fig. 1e). Induced cell death was not due to strain-specific differences in the proportion of infected cells, as shown by the comparable levels of viral nucleoprotein (NP) expression (Fig. 1c, f). These differences in cell death induction by infection with either a seasonal or pandemic IAV were confirmed by both TUNEL and lactate dehydrogenase release assays (Fig. 1g). Thus, RNA degradation correlated with induction of cell death in seasonal IAV-infected DC, with no cell death seen after pandemic DC infection.

**Seasonal IAV-induced cell death depends on viral RNA**. Programmed cell death may involve either extrinsic (receptor-mediated) or intrinsic signaling pathways[24]. To evaluate the role of extrinsic pathways in seasonal IAV-induced cell death, we measured the expression of cytokines including those known to

be involved in cell death induction[24–30] and examined the potential paracrine-signaling effects from seasonal IAV-infected DC on cell death induction in uninfected DC using a transwell culture system. Cytokine levels were generally low and in several cases below the limit of detection for the assay. Overall, levels were comparable between seasonal and pandemic IAV infection, with the exception of IFNβ. While IFNβ was markedly induced by seasonal NC/99, it was weakly induced by pandemic Cal/09 and by the other seasonal strain Tx/91 (Fig. 2a). Thus, the induction of IFNβ is not associated with IAV-induced DC death. Moreover, the absolute IFNβ levels observed were low compared to the levels detected following infection with Newcastle disease virus[31], which is a model system for immune activation[32]. The levels of TNFα and IP-10 tended to be higher following pandemic IAV infection than following cell death-inducing seasonal IAV infection. As the increases in both cytokines were seen with the pandemic virus, these results help to exclude the possibility that paracrine

signaling contributes to the cell death induced by seasonal virus infection. As a further exclusion of this possibility, we show that the presence of a TNFα antagonist or of neutralizing antibodies against TRAIL or FASL did not reduce NC/99-induced cell death (Supplementary Fig. 2), thus supporting the conclusion that external signaling does not play a significant role in cell death induction by seasonal IAV infection. The transwell experiment showed that secreted factors from NC/99-infected DC did not induce cell death in uninfected DC (Fig. 2b). Conversely, pretreatment of NC/99-infected DC with the RNA polymerase II and viral RNA-dependent RNA polymerase (RdRp) inhibitor actinomycin D suppressed cell death (Fig. 2c). Additionally, UV irradiation of NC/99 prior to DC infection significantly reduced cell death (Fig. 2d) in proportion to the reduction in virus infectivity (Fig. 2d, inset). Taken together, these results indicate that cell death induction involves cell-intrinsic mechanisms and likely requires virus RNA replication.

**NC/99-induced cell death is RIPK3–MLKL-caspase-mediated**. IAV infection has been associated with modulation of caspase-dependent apoptosis in several cell types[12]. Unlike apoptosis, necroptosis is independent of caspase activity. While RIPK3 and its substrate, mixed lineage kinase domain-like protein (MLKL)[33] are necessary to elicit necroptosis, RIPK1 is dispensable.

We sought to delineate the intracellular cell death pathways induced by seasonal IAV with pharmaceutical inhibitors and with CRISPR genetic ablation studies. Inhibition of RIPK3 kinase activity by either GSK872 or GSK840 dramatically decreased NC/99-induced cell death, provided caspases were concomitantly blocked by the pan-caspase inhibitor Z-VAD (Fig. 3a). Genetic ablation of RIPK3 alone or of both MLKL and FADD dramatically reduced NC/99-induced cell death in DC, supporting the idea that combined blockade of apoptosis and necroptosis signaling downstream of RIPK3, but not either pathway alone, is required for preventing IAV-induced cell death[34] (Fig. 3b, Supplementary Fig. 3). While NSA was identified as an MLKL inhibitor[35], the greater effect of NSA than of genetic ablation of MLKL on IAV-induced cell death suggests that in these cells, NSA also interferes with RIPK3-mediated caspase activation. The lack of effect of ablating RIPK1 or blocking its function by necrostatin-1 (Nec-1) further supports a specific RIPK3-dependent cell death mechanism[33, 36]. Infection with seasonal NC/99 IAV at two different multiplicities of infection (MOIs) significantly increased both MLKL phosphorylation and caspase 8 cleavage when compared to infection with Cal/09, further supporting the concept of dual activation of the apoptotic and necroptotic pathways by NC/99 (Fig. 3c, d and Supplementary Fig. 4)[19]. In contrast, infection with pandemic Cal/09 IAV did not significantly induce MLKL phosphorylation. As a control, necroptosis was induced by treatment with the combination of TNFα, cycloheximide, and Z-VAD (TCZ; see ref. [19]), which stimulated MLKL phosphorylation, but not caspase 8 cleavage. Furthermore, we show that genetic ablation of DAI, which has been identified as a viral RNA sensor leading to RIPK3-driven cell death in mice[37], significantly reduced NC/99-induced cell death (Fig. 3b). In aggregate, these results provide evidence that NC/99 infection activates parallel pathways downstream of the sensor protein DAI, the RIPK3–MLKL-dependent necroptosis pathway, as well as the RIPK3-caspase 8-mediated apoptosis pathway. As shown further below, activation of apoptosis in DC is nonimmunogenic, whereas the cell death induced by seasonal IAV infection is immunogenic. Thus, while seasonal virus activates both MLKL and caspase 8, the DC response to seasonal IAV infection is characteristic of an immunogenic

RIPK3–MLKL-mediated necroptosis, but not of a nonimmunogenic caspase-mediated cell death.

**Pandemic IAV inhibits RIPK3-mediated cell death**. We wondered if the absence of cell death in pandemic IAV-infected DC could result from an inhibitory mechanism. To address this question, we set out to determine the effect of pandemic Cal/09 infection on RIPK3-induced cell death in DC. We first tested whether transfection with the viral RNA mimetic poly I:C induced a RIPK3-mediated cell death in DC similar to that caused by seasonal IAV infection. Poly I:C was previously reported to induce necroptosis in macrophages[38]. DC transfection with increasing amounts of poly I:C led to a dose-dependent increase in cell death (Fig. 4a). While poly I:C-induced cell death was substantially reduced by the MLKL inhibitor NSA, it was not significantly altered by the RIPK1 inhibitor Nec-1 (Fig. 4b and Supplementary Fig. 5a). These results were consistent with induction of an MLKL-dependent, RIPK1-independent cell death pathway in infected DC. We next examined the effect of Cal/09 infection on poly I:C-induced cell death. The level of poly I:C-induced cell death was significantly reduced by Cal/09 infection (Fig. 4c), and the extent of this inhibition was proportional to the MOI (Fig. 4d), indicating that the pandemic virus suppressed DC death. Transfection efficiency of poly I:C was unaffected by either pandemic or seasonal IAV infection (Fig. 4e). We also tested whether poly I:C-induced cell death or the effect of pandemic virus on this cell death involved caspase signaling. Treatment with pan-caspase inhibitor Z-VAD had no effect on either poly I:C-induced cell death (indicating it was not caspase dependent) or on the inhibitory effect of Cal/09 on this RIPK3-dependent cell death (Supplementary Fig. 5b). Overall, these data support the formulation that the pandemic Cal/09 strain inhibits the development of RIPK3–MLKL-mediated necroptosis.

**RIPK3-mediated cell death activates T cell proliferation**. Necroptosis releases immunogenic DAMPs[39]. Cells undergoing necroptosis, unlike cells undergoing apoptosis, have been shown to activate neighboring DC to facilitate T cell activation and antigen cross-presentation[40, 41]. We sought to evaluate the influence of virus-induced DC necroptosis on T cell activation. Investigating the effect of necroptosis on the antigen-presenting capacity of uninfected DC would require the use of syngeneic T cells[42]. However, this is impractical in human primary cells due to the variability in memory T cells specific to different IAVs, especially the recent Cal/09 strain. Instead, we analyzed the impact of virus-induced DC necroptosis on the capacity of naive DC to cause allospecific stimulation of T cells[43], thus relying on superantigen TCR activation by MHC class I molecules, which is typical of a graft-vs.-host reaction[44, 45].

To assess the impact of virus-induced DC necroptosis on T cell activation, we used the following experimental approach: DC were pretreated or not with an MLKL inhibitor (NSA), infected with NC/99 or transfected with poly I:C, and cocultured with naive DC. The mixed DC were then cocultured with allogeneic T cells, and T cell proliferation was measured as the fluorescence decrease of carboxyfluorescein succinimidyl ester (CFSE)-labeled T cells using high resolution imaging flow cytometry. Stimulation of T cell proliferation using anti-CD3/CD28-coated beads served as a positive control. NSA pretreatment nearly abolished the induction of T cell proliferation by cocultures of naive DC with either NC/99-infected or poly I:C-transfected DC (Fig. 4f, g). Inducing apoptotic DC cell death using staurosporine or infecting DC with the pandemic Cal/09 virus did not cause naive DC to induce robust T cell activation (Supplementary Fig. 6). These results show that the dominant effect of seasonal IAV infection or

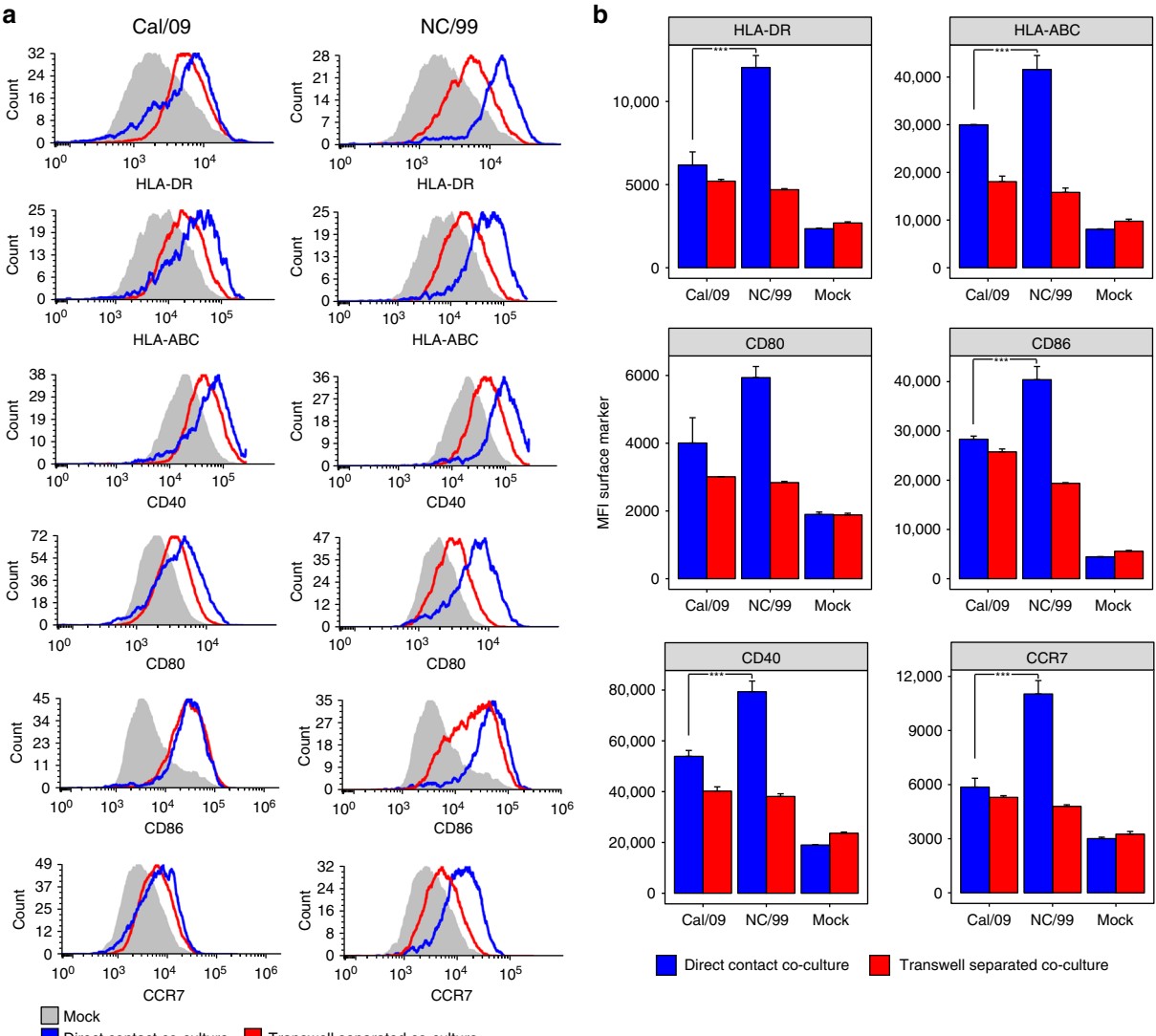

**Fig. 5** Analysis of maturation marker expression in naive DC cocultured with virus-infected DC either directly or in a transwell system. **a**, **b** Naive DC were cocultured directly (direct contact coculture) with either Cal/09- or NC/99-infected DC, at a ratio of 1:1. Alternatively, naive DC were cocultured in a transwell system (transwell-separated coculture) with either Cal/09-infected or NC/99-infected DC across the membrane at a ratio of 1:1. Cells were stained 18 h after infection for HLA-DR, HLA-ABC, CD40, CD80, CD86, CCR7, and NP, and analyzed by flow cytometry. Maturation marker expression was measured in NP-negative cells (Supplementary Fig. 7) in order to distinguish between infected and uninfected cells in the direct contact coculture. Mock-infected DC served as a negative control. **a** shows representative flow cytometry plots; **b** is a summary plot. Data are representative of three independent experiments using cells from different donors. $n = 3$ for experimental groups and for controls. ***$p < 0.001$, ANOVA followed by Tukey's HSD test, with an additional Bonferroni multiple-testing correction. Values shown are median ± s.e.m

poly I:C transfection in DC is a RIPK3-dependent immunogenic cell death that activates naive DC, thereby enhancing T cell activation.

As one reason for increased activation of T cells could be an increased expression of DC maturation markers, we assessed the activation of naive DC by DC undergoing cell death. For this purpose, we cocultured NC/99-infected or Cal/09-infected DC with uninfected DC, and analyzed the expression of cell surface maturation markers (CD40, CD80, CD86, CCR7, HLA-DR, and HLA-ABC) in the uninfected DC. Transwell experiments were conducted in parallel to assess potential paracrine-signaling effects on naive DC. While we observed a dramatic upregulation of DC maturation markers when NC/99-infected DC were in direct cell-to-cell contact with uninfected DC, this upregulation was much more modest when NC/99-infected DC were separated from uninfected DC by a transwell membrane (Fig. 5, Supplementary Fig. 7). There was little difference in the effects of

pandemic Cal/09 vs. seasonal NC/99 infection on maturation marker expression in uninfected DC separated from the infected cells by a transwell membrane. In contrast, direct contact with NC/99-infected cells caused a significant increase in nearly all maturation markers studied relative to direct contact with Cal/09-infected cells. Overall, these results suggest that direct cell-to-cell interactions between NC/99-induced dying DC and uninfected DC significantly contribute to activation of the uninfected DC, thus associated with an increase in MHC, co-stimulatory molecules, and CCR7 homing signaling.

**Viral genomic segment HA mediates necroptosis inhibition.** To identify the viral gene(s) that give pandemic IAVs the ability to suppress DC necroptosis, we studied recombinant Cal/09 viruses, in which segments encoding the major immune antagonist (NS1), the polymerase complex (PB2, PB1, and PA),

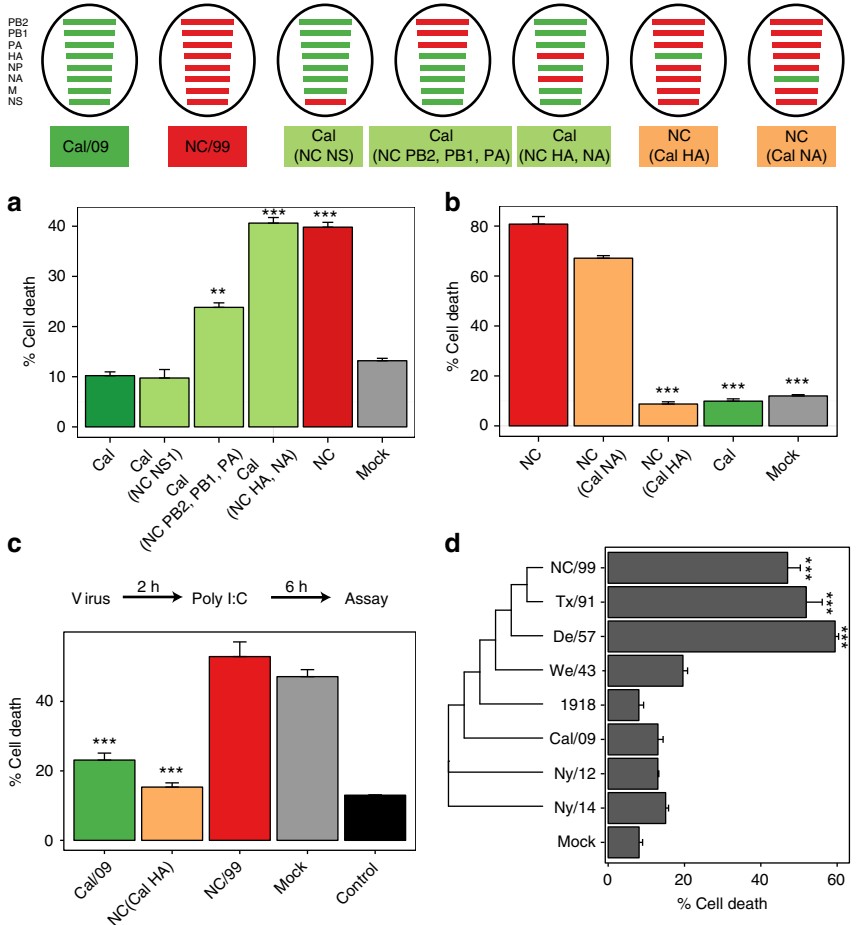

**Fig. 6** The pandemic HA viral segment mediates cell death inhibition. Top, Schematic of wild-type and recombinant IAVs. The genomic segments of pandemic Cal/09 and seasonal NC/99 IAV are indicated with green and red lines, respectively. **a**, **b** DC were either mock-infected or infected with various IAV constructs for 8 h. **c** DC were either mock-infected or infected with various IAV constructs for 2 h, and then transfected with poly I:C for 6 h. **d** Phylogenetic analysis showing HA sequence similarity in cell death-inhibiting or cell death-inducing IAV strains. **a–d** Percentage of cell death was determined by imaging flow cytometry. Data are representative of three independent experiments using cells from different donors. $n = 3$ for experimental groups and for controls. ***$p < 0.001$, ANOVA followed by Tukey's HSD test. Values shown are median ± s.e.m

or the surface glycoproteins (HA, NA) were substituted for their seasonal NC/99 counterparts (Fig. 6, schematic). DC infection with the recombinant Cal/09 containing the surface glycoproteins (HA, NA) from NC/99 restored a level of cell death comparable to that caused by infection with the parent NC/99 (Fig. 6a). Conversely, infection of DC with the recombinant NC/99 containing the HA segment from Cal/09 suppressed cell death (Fig. 6b), implying that the Cal/09 HA segment alone was necessary and sufficient to mediate cell death inhibition. It is noteworthy that all native and recombinant viruses were delivered at a comparable infectivity (with an MOI of 1), such that around 60% of DC expressed viral nucleoprotein NP (Supplementary Fig. 8a, b). Hence, the HA sequence-specific suppression of necroptosis is not the result of differences in infectivity. DC infection with the recombinant NC/99 background virus containing the Cal/09 HA segment caused a reduction in poly I:C-induced cell death comparable to infection with the parent Cal/09 virus (Fig. 6c), thus establishing the role of the pandemic HA segment in cell death suppression.

We next examined whether the HA surface protein itself had a repressive effect on DC necroptosis. DC transfected with poly I:C were exposed to either soluble HA from Cal/09 or virus-like particles (VLPs) bearing HA from Cal/09. Neither soluble HA nor the VLPs had a significant impact on poly I:C-induced cell death

(Supplementary Fig. 5c, d), thus suggesting that direct receptor binding is not involved in the inhibitory effect of HA. These results are consistent with the data in Fig. 2c, indicating that NC/99-induced cell death requires virus RNA replication. The fact that inhibition of cell death is observed following at least a 2-h infection period (Fig. 4c), further supports the idea that de novo RNA or protein synthesis of HA may be necessary for this inhibitory activity.

One cannot accurately model the systemic effects of the different IAV strains in vivo in mice for the following reasons: (i) mouse DC show low infectivity by the wild-type IAV strains studied, (ii) mouse DC show species-specific differences in infectivity by these IAV strains, and (iii) mouse vs. human epithelial cells may exhibit distinct cell death responses to IAV infection[22, 34]. Consequently, we performed correlative natural human experiments by comparing HA segment evolution with the capacity of an IAV strain to cause human DC death. Notably, during a century of virus–host adaptation, the most phylogenetically similar HA segments from different H1N1 virus strains were also the most similar in their capacity to either induce or inhibit DC death, with the emergence and reemergence of an HA-mediated suppression of necroptosis seen in the two H1N1 strains NY/12 and NY/14 derived from the original zoonotic pandemic 2009 H1N1 strain (Fig. 6d).

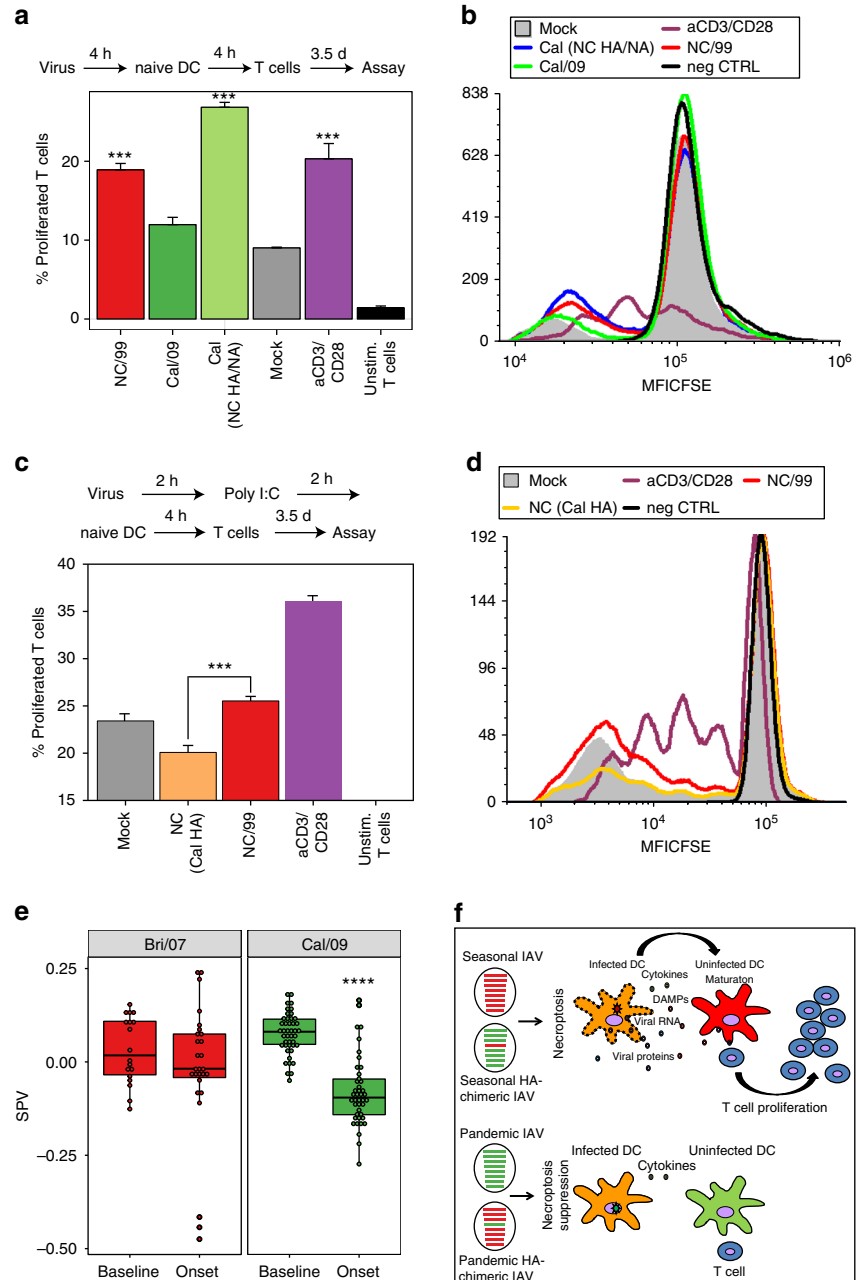

**Fig. 7** Pandemic IAV causes reduced T cell proliferation. **a**, **b** DC were either mock-infected or infected with the indicated IAVs for 4 h, cocultured with naive autologous DC for 4 h, and cocultured with CFSE-labeled allogeneic T cells for 3.5 days. **c**, **d** DC were either mock-infected or infected with the indicated IAVs for 2 h, transfected with poly I:C (4 μg/ml) for 2 h, cocultured with naive autologous DC for 4 h, and then cocultured with CFSE-labeled allogeneic T cells for 3.5 days. Percentage of proliferated T cells was determined by flow cytometry. T cells incubated with anti-CD3/CD28-coated beads served as a positive control. **a**, **c** are summary plots; **b**, **d** are representative flow cytometry plots from a replicate. **e** Blood CD8[+] T cell surrogate proportion variables (SPV) following symptomatic seasonal or pandemic IAV infection in humans were determined computationally. For the seasonal Bri/07 infection study, samples from all 9 subjects at two time points before infection (Baseline) were analyzed together; likewise, samples from nine subjects at 45.5, 53, and 60 h after infection (near the peak of influenza-induced symptoms; Onset), were analyzed jointly. Single dots represent individual data points. The three low-value outlier points post infection are from the same subject. Whether the outliers are excluded from the joint analysis, or data at each time point are analyzed separately, the results remain the same as in the joint analysis. The Cal/09 data represent analysis of a single preinfection sample and a single symptom onset sample coming from 45 subjects infected with the influenza virus. **f** Schematic illustrating the distinct effects of seasonal IAV and seasonal HA–chimeric IAV, as compared to pandemic IAV and pandemic HA–chimeric IAV, on the induction of necroptosis in infected DC, the subsequent maturation of uninfected DC, and its impact on T cell proliferation/activation. DAMPs, danger-associated molecular patterns. **a–e** Data are representative of three independent experiments using cells from different donors. $n = 3$ for experimental groups and for controls. ***$p < 0.001$, ANOVA followed by Tukey's HSD test, with an additional Bonferroni multiple-testing correction for summarized data; ****$p < 10^{-13}$, F-test. Values shown are median ± s.e.m

**Pandemic IAV reduces T cell proliferation in humans**. To further investigate the influence of pandemic IAVs on T cell response, we compared the capacity of pandemic vs. seasonal IAV-infected DC to stimulate allogeneic T cell proliferation. DC infected with pandemic Cal/09 IAV were inefficient at causing uninfected DC to stimulate T cell proliferation (Fig. 7a, b). In contrast, DC infected with either seasonal NC/99 virus or recombinant Cal/09 background virus containing the NC/99 HA segment were potent enhancers of T cell proliferation by uninfected cocultured DC. Furthermore, DC infection with the recombinant NC/99 background virus containing the Cal/09 HA segment, followed by poly I:C-treatment and coculture with naive DC elicited a significant decrease in T cell proliferation (Fig. 7c, d). These results support the formulation that the pandemic HA segment encodes a novel immune subversion effect that can undermine the adaptive immune response by inhibiting necroptosis.

To compare differences in T cell response to infection by pandemic vs. seasonal H1N1 IAVs in vivo, we used public human peripheral blood-based gene expression data to computationally determine[46] the proportion of circulating CD8+ T cells in individuals infected with either a pandemic Cal/09[47] or a seasonal Brisbane 2007 strain (Bri/07)[48]. The expression data were well resolved into relative cell-type proportion inferences (Supplementary Fig. 8c, d). At comparable time points after symptom development (onset), the circulating T cell proportion was reduced from baseline following infection with the pandemic strain (Cal/09) but was unchanged after infection with the seasonal strain (Bri/07; Fig. 7e). The levels of circulating T cells result from the effects of infection on the rate of T cell efflux from the circulation (homing to the site of infection) and the rate of new T cell entry into the circulation (related to T cell production). While further study of the underlying mechanism is warranted, the reduced T cell levels that we found after pandemic infection but not after seasonal infection in humans are consistent with the hypothesis that the lower DC immunogenic cell death following pandemic virus infection leads to a lower rate of T cell proliferation (Discussion section).

## Discussion

We find that seasonal IAV infection induces RIPK3-dependent immunogenic death of human DC, whereas pandemic IAVs suppress such death. We identify the pandemic HA segment as conferring pandemic IAV strains the capacity to specifically suppress this cell death. DC death would be expected to lead to the release of DAMPs, the activation of uninfected DC, and the stimulation of T cell activation and the adaptive immune response[40, 41](for review, see also ref. [49]). Consonant with this formulation, we show that DC infected with seasonal IAV strains, which induce death, can promote the capacity of uninfected DC to induce proliferation of allogeneic T cells (Fig. 7f). In contrast, DC infected with the pandemic viruses, which suppress death, do not have this effect on uninfected DC and on T cells. Furthermore, pandemic IAV infection in humans in vivo is shown to be associated with reduced levels of circulating T cells in comparison to seasonal IAV infection, an observation that is consistent with the lower T cell proliferation detected in vitro following pandemic IAV infection.

We find that the HA genomic segment mediates inhibition of the cell death pathway by pandemic IAVs. The IAV strains inducing cell death (seasonal NC/99, Tx/91, and De/57; Fig. 6d) have a phylogenetically similar HA segment, while strains that inhibit cell death (pandemic 1918, Cal/09, and NY/12 and NY/14) share a distinct pattern of genetic similarities in HA. Supporting our observations, previous phylogenetic studies showed that the

emergence of the 1918 H1N1 pandemic occurred with the introduction and effective adaptation of a novel HA subtype to humans, thus causing antigenic shift[50]. These analyses also revealed that some of the genes of the H1N1 seasonal lineage arose from the 1918 pandemic virus, and other genes from other circulating IAV strains. Similarly, pandemic H1N1/2009 emerged in humans from swine when the HA gene exhibited enough genetic and antigenic variation between the emergent strains and circulating seasonal viruses[51]. Thus, these genetic differences between pandemic and seasonal HA subtypes may account, at least in part, for their distinctive effects on cell death.

Several influenza viral proteins have been implicated in the manipulation of the apoptotic cell death pathway. NS1 has an antiapoptotic function, as it is an efficient IFNα/β antagonist, and type I IFNs potentiate IAV-induced apoptosis[52, 53]. M2 interferes with macroautophagy, thus favoring apoptosis[54]. While PB1-F2 was originally described as proapoptotic[55], induction of cell death has been reported in specific pandemic IAV strains, where it contributes to their virulence, thus highlighting strain-dependent differences[56]. Similarly, the anti-necroptotic role of the HA segment in pandemic IAV strains presumably contributes to their high pathogenicity. IAV infection starts with the binding of membrane-associated HA to sialic acid-containing receptors on the host cell membrane, which facilitates the virus entry into the cell[57]. HA then mediates fusion of the viral and endosomal membranes, ultimately leading to viral replication in the nucleus (for review, see ref. [58]). One potential candidate for IAV sensing and triggering necroptosis in DC is DAI (ZBP1/DLM-1), which has been recently implicated as a sensor for IAV RNA or protein and shown to activate RIPK3-dependent cell death in murine fibroblasts and lung epithelial cells[37, 59]. Our data showing significant reduction of NC/99-induced cell death by DAI genetic ablation support this idea (Fig. 3b). One possibility is that newly synthesized HA protein or HA RNA from pandemic IAV interferes with DAI, thereby disrupting the signaling cascade leading to necroptosis. Another possible role for HA is its interfering with ESCRT-III, which can hinder MLKL-induced disruption of the plasma membrane[60].

Using coculture and transwell experiments, we find that necroptotic seasonal IAV-infected DC are a potent stimulus for initiating DC maturation (Fig. 5). This maturation effect is heightened by direct contact between the infected and uninfected DC, while it is much more moderate when the cells are separated by a cytokine-permeable transwell membrane. Nonetheless, additional mechanisms may contribute to the increase in T cell proliferation following seasonal IAV infection. For example, it is possible that NC/99-infected DC may directly activate T cells (direct priming) more effectively than Cal/09-infected DC. In vivo, necroptotic DC may alert other local antigen-presenting cells (e.g., macrophages) via the release of DAMPs, rendering those cells stimulatory to the adaptive immune system.

Using allogeneic T cells to provide Signal 1 for T cell activation by DC, we find that when the naive DC are exposed to these dying DC, they become potent inducers of T cell proliferation. These results are consistent with the hypothesis that pandemic viruses, by suppressing death (especially necroptosis) and the transfer of danger signals and viral epitopes to uninfected DC, reduce T cell activation and the early evolving adaptive immune response to infection (Fig. 7f).

In vivo data are necessary to explore the relevance of this hypothesis to the immune response to IAV infection in humans. When specifically comparing the effects of human IAV strains on the immune response in vivo, it is important to consider that the differing mechanisms of virus–host immune cell interaction used by specific IAV strains are unlikely to be faithfully preserved when studying hosts such as rodents, which the virus did not

evolve to infect. We have previously found, for example, that unlike human DC, mouse DC are highly resistant to nonadapted wild-type IAV infection[22]. In order to obtain data from human infection to compare to the results we obtained in human cells in vitro, we used public global transcriptome data following human pandemic and seasonal IAV infections to infer the proportion of circulating T cells. We find that the pandemic infection leads to a significantly lower level of circulating CD8 T cells than the seasonal infection. The results from human infection in vivo, while significant, are based on relatively small numbers of infected patients. Data from a larger cohort of experimentally infected or well-documented IAV infections in humans that could replicate these results are not currently available. While our findings are consonant with the formulation that the pandemic virus infection leads to less T cell proliferation, as suggested by the in vitro studies, the mechanisms underlying the lower levels of circulating T cells after IAV infection in vivo are complex. The levels of circulating T cells result from the net T cell flux, i.e., the rate of T cells entering the circulating pool from tissues minus the rate of T cells leaving the circulating pool. T cells recirculate among blood and secondary lymphoid organs (e.g., spleen, lymph nodes, and Peyer's patches), up until they encounter their cognate antigen following microbial infection. They then differentiate into memory/effector T cells that have the ability to migrate to lymphoid organs, as well as extralymphoid tissues and inflammatory sites (e.g., intestinal lamina propria, pulmonary interstitium; for review, see ref. [61]). Further clinical study is needed to replicate these findings and to determine whether these results on circulating T cells indeed stem from a differential T cell production rate after seasonal vs. pandemic human IAV infection.

The capacity for IAV infection to induce RIPK3-driven cell death has been identified in a mouse model using a mouse-adapted strain. Mouse infection with the PR8 IAV strain activates parallel pathways of necroptosis and apoptosis downstream of RIPK3 in fibroblasts and airway epithelial cells, both in vitro and in vivo[19]. We now demonstrate in human DC that the capacity to induce RIPK3-mediated cell death is IAV-strain specific. While the human seasonal viruses studied induce such death, the pandemic H1N1 viruses have the capacity to suppress cell death. Based on these findings, we propose that this pandemic H1N1 virus suppression of programmed necrosis is a novel mechanism for interfering with the development of the host immune response and may contribute to the unique pathogenicity of the pandemic viruses.

## Methods

**Dendritic cell preparation**. All human subject research studies were reviewed and approved by the IRB of the Icahn School of Medicine at Mount Sinai (ISMMS). Informed consent was obtained from nonanonymous donors. Monocyte-derived DC were obtained from buffy coats from human blood donors following a standard protocol[10]. Briefly, PBMC were isolated from buffy coats by Ficoll density gradient centrifugation, and CD14+ monocytes were immunomagnetically purified, which was followed by differentiation into DC during a 5-day incubation in DC growth medium containing 500 U/ml hGM-CSF (Preprotech) and 1000 U/ml hIL-4 (Preprotech). All experiments were replicated using cells obtained from different donors. Monocytes were phenotyped for various DC markers before and after differentiation into DC (Supplementary Fig. 1).

**Virus preparation and viral infection**. The human isolates of H1N1 influenza A viruses A/Brevig Mission/1/1918 (1918), A/California/4/2009 (Cal/09), A/New Caledonia/20/1999 (NC), A/Texas/36/1991 (TX), A/Denver/57 (De/57), A/Weiss/43 (We/43), A/New York/46/2012 (NY/12), and A/New York/WC-LVD-14–044/2014 (NY/14) were propagated in specific pathogen-free embryonated hen's eggs[62] (Charles River Laboratories). All experiments involving the 1918 strain were conducted under high-containment (enhanced biosafety level 3 [BSL3]) laboratory conditions in accordance with guidelines of the ISMMS Institutional Biosafety Office, the National Institutes of Health, and the Centers for Disease Control and Prevention. Recombinant influenza viruses were developed by reverse genetics techniques[63]. Infectious titers of influenza viruses were determined by standard plaque assay on Madin–Darby canine kidney (MDCK) epithelial cells. Prior to each

experiment, the infectivity of each virus preparation in DC was measured by influenza nucleoprotein (NP) staining and its titer was adjusted so that each strain infects approximately 60% of the DC obtained from each of 6 anonymous donors. For infection, virus stocks were diluted in serum-free medium and added directly onto pelleted DC at a multiplicity of infection (MOI) of 1. DC were infected in triplicates. After infection in RPMI medium at 37 °C for 10 min, cells were centrifuged to remove the viral inoculation medium, and resuspended in DC growth medium.

**Dendritic cell perturbations**. DC were transfected with poly I:C (Invivogen, tlrl-piclv; typically used at a concentration of 4 μg/ml) in combination with a transfecting agent, Lyovec (Invivogen), following the manufacturer's instructions. DC were treated with the following small chemical inhibitors: necrosulfonamide at 15 μM (EMD Millipore), GSK840 and GSK872 at 5 μM (Abious), SYN-1215 at 50 nM (AdipoGen), actinomycin D at 2 μM, and necrostatin at 30 μM (Sigma Aldrich). DC paracrine signaling was studied in a transwell system (Millicell-24 Cell Culture Insert Plate, polycarbonate, 0.4 μm, EMD Millipore). Z-VAD-FMK (20 μM alone; 50 μM in the TCZ combination) and cycloheximide (2 μg/ml) were purchased from BD Biosciences. TNFα (0.1 μg/ml) was from Symansis.

**Imaging flow cytometry**. For cell death analysis and determination of infectivity, cells were fixed with 1% paraformaldehyde (Electron Microscopy Science), permeabilized with methanol (Sigma), washed in PBS, and stained with influenza NP-specific antibodies (Abcam, ab20921; used at a 1:100 dilution) and Hoechst 33342 (Invitrogen; used at a 1:1000 dilution) to label nuclei. Single-cell images were acquired using the ISX Imaging flow cytometer (Amnis). The percentage of cell death among DC was inferred from the number of cells with fragmented nuclei and altered cell morphology. Nuclear fragmentation was assayed by determining the area of the pixels containing the upper 30% intensity in the frequency of the nuclear dye. Cell morphology was assessed by quantifying brightfield contrast.

**TUNEL and lactate dehydrogenase cytotoxicity assays**. To quantify cell death, the APO-BrdU terminal deoxynucleotidyl transferase dUTP nick end labeling (TUNEL) assay kit from ThermoFisher Scientific was used in combination with a flow cytometer for fluorescence detection, following the manufacturer's guidelines. The lactate dehydrogenase (LDH) cytotoxicity assay kit manufactured by Pierce was used to measure cell death, according to the manufacturer's guidelines. LDH measurements were carried out on a microtiter plate reader. The percentage of maximal LDH release was calculated relative to the measurement obtained from lysed cells.

**Cytokine measurements**. The levels of all cytokines, except IFNβ, secreted upon DC infection with either pandemic or seasonal IAV strains were measured using a multiplex ELISA kit, following the manufacturer's recommendations (EMD Millipore). IFNβ was measured using a conventional ELISA kit (PBL Interferon Source).

**Genetic ablation**. Guide RNAs were constructed using the GeneArt Precision gRNA Synthesis Kit (Life Technologies). Guide RNAs were mixed with GeneArt™ Platinum™ Cas9 Nuclease to form ribonucleoproteins (Invitrogen). Monocytes were transfected with those ribonucleoproteins via electroporation using the Neon Transfection system (Life Technologies). Electroporated monocytes were cultured into DC in the presence of IL-4 and GMCSF, and then infected with either mock or IAV. Cell death and genetic ablation were assayed by imaging flow cytometry through quantification of nuclear fragmentation and assessment of protein expression of the ablated gene, respectively (Supplementary Fig. 3). The sequences of the forward and reverse PCR primers used for guide RNA synthesis were IVT-FADD-gRNA-T1-fwd, 5′-TAA TAC GAC TCA CTA TAG AGC TCA AGT TCC TAT G-3′; IVT-FADD-gRNA-T1-rev, 5′-TTC TAG CTC TAA AAC GAG GCA TAG GAA CTT GAG C-3′; IVT-FADD-gRNA-T3-fwd, 5′-TAA TAC GAC TCA CTA TAG TGA CGT TAA ATG CTG C-3′; IVT-FADD-gRNA-T3-rev, 5′-TTC TAG CTC TAA AAC GTG TGC AGC ATT TAA CGT C-3′; IVT-MLKL-gRNA-T1-fwd, 5′-TAA TAC GAC TCA CTA TAG TTG AAG CAT ATT ATC A-3′; IVT-MLKL-gRNA-T1-rev, 5′-TTC TAG CTC TAA AAC AGG GTG ATA ATA TGC TTC A-3′; IVT-MLKL-gRNA-T3-fwd, 5′-TAA TAC GAC TCA CTA TAG CAG GAC GCT CCT GGG C-3′; IVT-MLKL-gRNA-T3-rev, 5′-TTC TAG CTC TAA AAC ATA AGC CAA GGA GCG TCC T-3′; IVT-RIPK1-gRNA-T1-fwd, 5′-TAA TAC GAC TCA CTA TAG CGG CTT TCA GCA CGT G-3′; IVT-RIPK1-gRNA-T1-rev, 5′-TTC TAG CTC TAA AAC GAT GCA CGT GCT GAA AGC C-3′; IVT-RIPK1-gRNA-T3-fwd, 5′-TAA TAC GAC TCA CTA TAG TGG GCG TCA TCA TAT A-3′; IVT-RIPK1-gRNA-T3-rev, 5′-TTC TAG CTC TAA AAC TTC CTC TAT GAT GAC GCC C-3′; IVT-ZBP1-gRNA-T1-fwd, 5′-TAA TAC GAC TCA CTA TAG ACT CCT TTT TCA TTC G-3′; IVT-ZBP1-gRNA-T1-rev, 5′-TTC TAG CTC TAA AAC CTA CCG AAT GAA AAA GGA G-3′; IVT-ZBP1-gRNA-T2-fwd, 5′-TAA TAC GAC TCA CTA TAG AGT CCT CTA CCG AAT G-3′; IVT-ZBP1-gRNA-T2-rev, 5′-TTC TAG CTC TAA AAC TTT TCA TTC GGT AG AGG AC-3′; IVT-RIPK3-gRNA-T2-fwd, 5′-TAA TAC GAC TCA CTA TAG CAG TGT TCC GGG CGC A-3′; IVT-RIPK3-gRNA-T2-rev, 5′-TTC TAG CTC TAA AAC ATG GTT GCG CCC GGA CAC T-3′; IVT-ZBP1-gRNA-T3-fwd, 5′-TAA TAC GAC

TCA CTA TAG AAT TCG TGC TGC CGC T-3′; IVT-RIPK3-gRNA-T3-rev, 5′-TTC TAG CTC TAA AAC TTC TAG GC CAG CAC GAA T-3′.

**Flow cytometry**. The following antibodies were used in flow cytometry experiments: anti-phosphorylated MLKL (Abcam, ab196436; used at a 1:50 dilution) labeled with Ax488 using the Zenon staining kit (Life Technologies), and anti-cleaved caspase 8 (BD Biosciences, 551244; used at a 1:100 dilution). DC were infected and fixed with paraformaldehyde (PFA), permeabilized with ice-cold methanol, and stained with antibodies.

**Allospecific stimulation of T cells**. DC were infected with IAV strains at a MOI of 1. Four hours after infection, DC were washed and cocultured with naive DC for another 4 h at a naive-to-infected DC ratio of 3:1. Allogeneic T cells labeled with carboxyfluorescein succinimidyl ester (CFSE; Invitrogen) were then added at a T cell-to-DC ratio of 2:1. T cell proliferation was assessed by measuring the corresponding decrease in cell fluorescence by flow cytometry. As a positive control, anti-CD3/CD28-coated beads (Life Technologies) were used to stimulate T cell proliferation.

**Phylogenetic analysis**. HA sequences from A/Brevig Mission/1/1918 (1918), A/California/4/2009 (Cal/09), A/New Caledonia/20/1999 (NC), A/Texas/36/1991 (TX), A/Denver/57 (De/57), A/Weiss/43 (We/43), A/New York/46/2012 (NY/12), and A/New York/WC-LVD-14-044/2014 (NY/14) were used for a phylogenetic analysis using the phyml algorithm (DOI: 10.1080/10635150390235520) with GTR as a run model within the flu database website (www.fludb.org). The actual tree graph was made in R using the package GGTREE[64].

**Computational estimate of CD8 T cell proportion in blood**. We analyzed public microarray blood expression data of subjects exposed to pandemic and seasonal H1N1 influenza challenges to computationally infer whether cell-type proportions in blood following the challenges differ from baseline. The pandemic Cal/09 infection data were obtained from a prospective study[48] that included microarray data (GEO GSE68310) for 45 individuals infected with the influenza H1N1 virus exclusively. The seasonal H1N1 A/Brisbane/59/2007 infection data were from an intranasal influenza inoculation study of healthy volunteers[47], which included microarray data (GEO GSE52428) for 9 individuals that became symptomatic and were determined to be microbiologically infected with the H1N1 strain.

For the GSE68310 data, the basic microarray data processing was performed, as described in the study, retaining all transcript probes in the microarray with detection $p \leq 0.05$ in at least 70% of the samples. The data were then quantile normalized and log-transformed. We compared the analysis of the baseline blood draw at the time of enrollment in the study and the first time point after subjects reported with influenza-like symptoms. For the GEO GSE52428 data, the normalized data were log-transformed, and probes maximally expressed at background level were removed. Data for both studies were further processed by selecting the probe with the highest average expression over the full time course for each gene. Finally, all genes that showed a low average expression and low variance, as defined by falling below the 20th quantile for both statistics, were removed.

To resolve the relative cell-type proportion changes pre infection and post infection, we applied the CellCODE algorithm to these microarray studies[46]. For the seasonal Bri/07 study, to increase the sample size used for inference of cell-type proportion changes, we pooled the data from two baseline blood draws (18 samples in total), one done at the time of enrollment and the other immediately prior to inoculation. We also pooled the data from three blood draws (27 samples in total) sampled at 45.5, 53, and 60 h post inoculation, which are comparable in time relative to symptom onset in the pandemic data.

The CellCODE algorithm has been demonstrated to provide accurate estimates of relative cell-type proportions in blood that are highly concordant with cell-type proportions determined experimentally[46]. The CellCODE approach estimates the relative differences in cell proportions directly from the molecular expression measurements, while relying on external information regarding which marker genes are likely to track cell-type abundance. We used the marker lists derived from the IRIS (Immune Response In Silico)[65] and DMAP (Differentiation Map) data sets[66]. In application to both IAV data sets (Supplementary Fig. 8c, d), as expected, markers for the same cell type correlate with each other, while markers for different cell types are uncorrelated, resulting in a block-like correlation structure that is captured by the CellCODE surrogate proportion variables.

**Statistical analysis**. For all in vitro data, similarity of variance between groups was compared using the Bartlett's test. Data were then tested for normality using the Shapiro–Wilks test. Data were analyzed with ANOVA using R[67], followed by pairwise comparisons using Tukey's "Honest Significant Difference" method and an additional Bonferroni multiple-testing correction for the summarized data. Plots depicting data were constructed using the R package ggplot2[68]. No statistical methods were used to predetermine sample size.

**Data availability**. The data sets generated during and/or analyzed during the current study are available from the corresponding author on reasonable request.

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

## Acknowledgements

This work was supported by PRIME (Program for Research on Immune Modeling and Experimentation), an NIAID-funded Modeling Immunity for Biodefense Center (Grant U19 AI117873), and was partly supported by CRIP (Center for Research on Influenza Pathogenesis), an NIAID-funded Center of Excellence in Influenza Research and Surveillance (CEIRS, contract number HHSN272201400008C), and U19 AI106754. We thank Yongchao Ge for his advice on statistical analysis, Richard Cadagan for technical assistance, and Florian Krammer for providing soluble HA. We are thankful to Kirsten St. George at Kirsten St. George Wadsworth Center for providing the NY/12 and NY/14 IAV strains.

## Author contributions

Experiments were designed by B.M.H., R.A.A., E.Z., S.B., A.G.-S., and S.C.S. Experiments were conducted and analyzed by B.M.H., R.A.A., E.Z. G.N., N.M., C.M.-R., R.F., J.P.I., I.R., and M.S. with supervision from S.B., A.G.-S., and S.C.S. The manuscript was drafted by B.M.H., R.A.A., H.P., and S.C.S. with contributions from all authors.

## Additional information

**Competing interests:** The authors declare no competing financial interests.

