## [Peer Review File · Nature Communications]

Reviewers' comments:

Reviewer #1 (Remarks to the Author):

A. Key Results

Infection of monocyte-derived DC with seasonal influenza virus strains causes cell death that is reduced in the presence of drugs that target the necroptosis pathway. HA is the viral protein responsible.

Infection of monocyte-derived DC with pandemic influenza virus strains reduces necroptosis resulting from transfection with poly IC.

Infection of monocyte-derived DC with seasonal influenza virus strains results in increased DC-mediated T cell proliferation. HA is the viral protein responsible.

B. Originality and Interest.

The manuscript is lacking in originality largely because the authors themselves have recently published: "Suppression of dendritic cell necroptosis by pandemic influenza virus" Hartmann et al, 2016 in Journal of Immunology. Currently, only the abstract is available however the abstract of this article describes the key findings of this current manuscript (seasonal, but not pandemic influenza induces DC necroptosis and HA is responsible).

In addition, the following findings have already been described, directly relevant to the major themes of the manuscript:

(i) differing immune responses, including altered T cell responses, to pandemic v seasonal influenza virus infection eg. Meunier et al. Plos One 2015

(ii) influenza virus induction of necroptosis eg. McComb et al. Cell Death Differentiation 2012; Rodrigue-Gervais et al. Cell Host Microbe 2014.

D. Appropriate use of Statistics.

Figure 2F is overexposed and unconvincing. The data demonstrating MLKL phosphorylation in particular has been over interpreted. This requires quantitation from independent experiments if statements are to be made about significant differences. The authors should also evaluate and comment on the levels of FADD/RIPK1/RIPK3 under the different infection conditions.

E. Conclusions.

Identifying necroptosis as the mode of cell death elicited by seasonal influenza infection relies on treatment of cells with drugs only, no genetic evidence is provided.

F. Suggested Improvements

How do the authors exclude that the partial protection from necroptosis that occurs following pandemic influenza infection is not due to reduced transfection with poly IC? What is the mechanism by which pandemic influenza-infected cells are protected from necroptosis? Note that in Figure 3A the amount of cell death inflicted by poly IC is

significantly lower compared to panels b and c.

It is not clear what the assays of T cell proliferation are measuring. More detail is required to interpret these outcomes. Are influenza infected DC providing antigen for naive DC or eliciting naive DC cytokine production? What are the source of T cells and/or their specificity?

HA is identified as the gene segment responsible for the enhanced death of DC infected with seasonal influenza. What is the mechanism responsible? Data in Figures 4d-f show a correlation with seasonal influenza virus infection or seasonal virus HA with the ability of infected DC to elicit increased T cell activation. Again, what is the mechanism responsible? There is no direct evidence linking these outcomes with necroptosis. Controls are required for the infectivity of the reassortment viruses.

It is not clear that monocyte-derived DC are a relevant cell type for infection with influenza virus infection. What is the evidence that these cells participate in the response to influenza virus infection in vivo? The manuscript would strongly benefit from data obtained in vivo.

G. References and H. Clarity and Context

The Introduction & Discussion are significantly lacking in detail. The manuscript requires more detail regarding what is already known about (i) dendritic cells and influenza A virus infection, (ii) immune outcomes associated with pandemic versus seasonal influenza A virus strains and (ii) the role of necroptosis in influenza A virus. There are relevant publications for each of these topics, none of which are cited.

Reviewer #2 (Remarks to the Author):

The submission by Hartmann et al described the consequences of pandemic H1N1 IAV infection in human DC. The authors present data showing that infection with seasonal IAV leads to rapid host RNA degradation, but infection with pandemic strains do not cause such a loss in RNA. This loss relates to seasonal (but not pandemic) IAV infection inducing DC death. The strength of the manuscript is the data in Figure 4, showing the importance of HA in the induction of necroptosis. This strength is outweighed by numerous weaknesses (described below). Of particular concern, the brevity of the Introduction and Discussion makes it hard for the reader to grasp the initial importance of the study, as well as the importance of the data presented. In addition, this submission lacks considerable novelty based on the recent paper published by Nogusa et al, which included as authors the first and senior authors of this manuscript. Collectively, these points decrease the overall enthusiasm for this manuscript.

Weaknesses:

1. The authors are using monocyte-derived DC in the experiments. There is no data presented showing the phenotype of the cells after the differentiation process.

2. The authors present data showing the DC death after seasonal IAV infection was via necroptosis and RIPK3-dependent. This data generally replicates the data recently published by Nogusa et al. in *Cell Host & Microbe*. Interestingly, the authors of this manuscript were co-authors on the Nogusa et al publication, which went into more extensive detail describing the mechanism of the IAV-induced necroptosis. Granted, the Nogusa et al paper did everything in mouse cells and used the common mouse-adapted IAV strains PR/8 and x-31 - while the submission by Hartmann et al uses human DC and human IAV strains. Regardless, the overall novelty of the manuscript under review is significantly decreased because of this other paper already published.
3. The T cell proliferation data in Figure 3d does not make sense. First, what is driving the T cell proliferation? There is no Ag provided. Second, while the authors include necrosulfonamide to block necroptosis, the inclusion of cells killed by some other mechanism (e.g., apoptosis) would have been a nice control. Third, the authors do not provide any data examining the phenotype of the uninfected bystander DC presumably being activated by DAMPs.
4. The data in Figure 4 is the most interesting, which identified HA as the viral component that induced necroptosis. The authors need to expand their investigation of the differences in the HA proteins from the different IAV used. At least, a better discussion of potential mechanisms defining the "HA effect" should have been done.
5. The Discussion is almost non-existent, and the text that is there is mostly a rehashing of the results section.

Reviewer #3 (Remarks to the Author):

In this manuscript Hartmann et al present data showing that pandemic influenza virus (IAV) strains prevent dendritic cell necroptosis as opposed to seasonal influenza strains, and that this inhibition is dependent on the hemagglutinin (HA) component of pandemic strains. Many aspects of the paper relating human IAV strains to host dendritic cell necroptosis (e.g. dependence on viral RNA/ replication and RIPK3 for IAV induced cell death) are replications of data already reported by Shoko et al. (2016) with mouse IAV strains upon infection of mouse fibroblasts, mouse epithelial cells and human epithelial cells, and although important, these are incremental. The main novelty of this study is the finding that HA of human pandemic IAV strains but not seasonal strains inhibits necroptosis, which is interesting, but several pieces of data need to be adequately worked up and additional mechanistic insight provided for publication in *Nature Communications*. Some sections of the manuscript notably the introduction and discussion need to be re-written.

Major comments:

- 1) The authors largely present their findings in a clear and succinct manner, however the introduction and discussion are too short. These sections need to be expanded. The introduction does not provide a sufficient review of the existing literature and needs to be developed into a coherent form that provides the reader a strong basis for comprehending the manuscript going forward. For instance what is the current understanding of the role of cell death in virus (IAV)-host interactions (e.g. how do viruses like IAV manipulate cell

death to perhaps promote infectivity and/or how does the host exploit cell death to its benefit)? What is our current understanding of the arms race between IAV and the host? Once a sound review is provided, the authors need to build on it to pose the question under investigation in the current manuscript. As the intro currently stands, it is left to the reader to deduce or else to extensively inspect the literature to understand a) what the question/hypothesis under investigation is, b) why it is novel and c) why was it posed in the light of current literature? Similarly, the discussion needs to integrate the authors' findings with current cell death / IAV literature and perhaps speculate on its importance, rather than serve as a short summary of the results.

2) Fig. 2d: Cell death in necrosulfonamide (MLKL inhibitor) or necrosulfonamide+ZVAD treated cells is approx. 50% lower than that in untreated or ZVAD alone conditions at baseline (i.e. in the absence of NC/99 virus infection). In the virus-infected condition, one sees the same 50% reduction in cell death between these treatments. Therefore the effective fold increase in cell death upon virus infection is ~2 fold irrespective of MLKL inhibition. From these data, one cannot convincingly conclude that the reduction seen with MLKL inhibitor is due to an active role for MLKL during NC/99 infection, and not a result of reduced responsiveness due to baseline status of cells treated with MLKL inhibitor. The effects of necrosulfonamide in Fig. 2e are more convincing, but it is unclear which panel represents real biology (2d or 2e). Please clarify and replace the data in Fig. 2d with an experiment devoid of significant baseline effects of the inhibitor alone.

3) The authors need to provide mechanistic insight into how/why DC necroptosis enhances T cell proliferation/adaptive immunity. Figure 3d contains some good data. The authors should determine how incubating DCs with necroptotic cells influences a) their ability to make cytokines, b) express co-stimulatory molecules and c) to present antigen, and how this affects the ability of T cells to make cytokines.

4) It is well known that small molecule inhibitors can have off target effects. Therefore, data from cells where RIPK3, MLKL and RIPK1 expression has been ablated with either siRNA or CRISPR/Cas9 are absolutely necessary to complement the inhibitor data (Figs. 1 and 2). Where available, additional inhibitors of RIPK3 and MLKL should be used.

5) HA is tightly linked to influenza entry into cells. Therefore, the data shown in Figure 4 could arise from differences in the infectivity of the recombinant viruses towards DCs. Demonstrating equal infectivity and viral replication by assaying for expression of NP, NS1 or another non-structural protein is absolutely essential for these data to be indicative of a role for HA in preventing DC necroptosis. Given that this is the main point of the paper, the authors should show that HA alone inhibits necroptosis. This can be done by transfecting purified HA or plasmids expressing HA from seasonal or pandemic IAVs into DCs and examining polyI:C-induced necroptosis in these cells.

6) Extended Data Fig. 3: Here one hardly sees any T cell proliferation with NC/99 infected DCs compared to mock infected DCs which is contrary to that shown in Fig. 4e. This figure needs to be changed/redone. Please also add representative CFSE FACS plots to Fig. 4e and Extended Data Fig. 3.

7) Fig. 4f: The seasonal influenza (Bri/07) data are only from 9 individuals and in 3 individuals (i.e. a third of the sample size) the CD8 T cell proportion totally crashes compared to the pre Bri/07 infection. Given the heterogeneity in the post Bri/07 condition, this is a sample size too small to draw any biologically meaningful conclusions. To be suitable for publication, the authors need to reproduce this trend by pooling data from other

seasonal IAV studies to get a larger sample size.

8) Why does CD8 T cell abundance in Cal/09 infected individuals fall below that in the pre infection condition? This seems unusual but the authors don't comment on it. Please explain.

Minor comments:

1) How was the plot in Fig. 1a generated? What is the intact nucleus score and how was it computed? Y-axis labels should be provided and the basis for assigning a specific score to a cell should be described in the methods or in the figure legend.

2) Insets in Figs. 1b and 2c should have their own designation within Figs. 1 and 2.

3) Concentrations of actinomycin D, all other inhibitors and poly I:C used in each instance should be specified in the methods section and figure legends.

4) Figure 2f and Page 5 (Section titled "NC/99-induced cell death is RIPK3-dependent"): The authors' claim that infection with the pandemic Cal/09 IAV strain did not significantly increase MLKL phosphorylation is contrary to what their western blot data in Fig. 2f show. The western blot clearly shows that following Cal/09 IAV infection there is an increase in MLKL phosphorylation at 8 hours compared to mock infection, albeit lesser than that seen with the NC/99 strain. The statement on Page 5 should therefore be reworded to accurately reflect this data.

5) Fig. 4d: How was the phylogenetic analysis done? Please provide details in the methods section.

6) Fig. 3c: How was this experiment done? Were cells pretreated with different MOIs of Cal/09 followed by Poly I:C transfection? Please fix the figure legend for this panel.

7) Extended Data Fig. 4 is not cited anywhere in the main text except the methods. Because this figure lays the foundation for the subsequent CD8 T cell analysis please discuss it in the main text.

8) Extended Figure 2a: Increase in MFI of pMLKL upon NC/99 infection is very modest, which isn't completely consistent with the much larger increase in pMLKL shown in Fig. 2f. Please explain. Please also show representative FACS plots for pMLKL.

9) Extended Figure 2b: What is the scale on the X-axis? A graph corresponding the flow plots (as in Extended Fig. 2a) should be shown. Please also state the number of replicates and repeats for data, and what the error bars represent in Extended Figs. 2 and 3.

10) Extended Figure 2c: Is the X-axis label correct? A concentration of mg/uL of Poly I:C sounds quite high.

11) When processing data for CellCODE analysis, can the authors explain why they select only the probe with the highest average expression for each gene (as mentioned in the methods section on Page 12)? Part of the advantage of having multiple probesets for each gene on microarray is to bypass any artifacts that may arise due to single probesets.

12) Page 8, Discussion: Please fix the following sentence "As transfection of the viral RNA mimetic Poly I:C by alone induces necroptosis (Extended Data Fig. 2c), the inhibition". Delete 'by'.

Response to Reviewers

Note: Reviewers' comments are in <<italics>> and authors' responses are in plain text.

Authors' Response Introduction: This manuscript was originally submitted as a short report, which severely limited the references cited, the introduction and the discussion. A number of the comments result from the omissions due to the terseness of this submission format. We have now expanded the introduction and discussion and improved the comprehensiveness of literature citations, explanations of assays utilized and interpretation of the results.

Reviewer I:

<<The manuscript is lacking in originality largely because the authors themselves have recently published: "Suppression of dendritic cell necroptosis by pandemic influenza virus" Hartmann et al, 2016 in Journal of Immunology. Currently, only the abstract is available however the abstract of this article describes the key findings of this current manuscript (seasonal, but not pandemic influenza induces DC necroptosis and HA is responsible).>>

Authors' Response: This abstract cited by the reviewer represents the abstract for a presentation on the topic of this paper at the Immunology 2016 meeting of the AAI. Meeting presentations of unpublished results are not prior publication.

<<In addition, the following findings have already been described, directly relevant to the major themes of the manuscript:

(i) differing immune responses, including altered T cell responses, to pandemic v seasonal influenza virus infection eg. Meunier et al. Plos One 2015>>

Authors' Response: The interesting Meunier *et al.* study, which we now cite in our introduction, shows the effects of mouse-adapted viruses on mouse mortality and immune response. It, however, does not address the differential effects of pandemic versus seasonal wild-type human viruses on RIPK3-dependent cell death outcomes in key immune cells, which is the subject of our study.

<<(ii) influenza virus induction of necroptosis eg. McComb et al. Cell Death Differentiation 2012>>

Authors' Response: We believe that the reviewer is referring to the following article:

<https://www.ncbi.nlm.nih.gov/pmc/articles/PMC3469059/>

However, this article does not address influenza-mediated RIPK3 cell death/necroptosis. This paper by McComb and coworkers studies the action of SMAC mimetics on macrophages and its effect on parasite infection, which is unconnected to the subject of our paper.

<<(iii) Rodrigue-Gervais et al. Cell Host Microbe 2014.>>

Authors' Response: The Rodrigue-Gervais *et al.* (Cell Host Microbe 2014 PMID 24439895) as well as the Nogusa *et al.* (Cell Host Microbe 2016 PMID 27321907) reports studies of mouse-adapted influenza virus infection in mice. Our paper describes blockade of RIPK3 necroptotic cell death by pandemic viruses in

HUMAN cells, a natural host of wild-type influenza A viruses Also, Rodrigue-Gervais *et al.* and Nogusa *et al.* do not study *differences* between seasonal and pandemic strains, nor do they identify a determinant (IAV HA gene segment) of cell death susceptibility that differentiates seasonal from pandemic 1N1 IAV. Both these studies are cited in the manuscript.

<<Figure 2F is overexposed and unconvincing. The data demonstrating MLKL phosphorylation in particular has been over interpreted. This requires quantitation from independent experiments if statements are to be made about significant differences. The authors should also evaluate and comment on the levels of FADD/RIPK1/RIPK3 under the different infection conditions.>>

Authors' Response: In order to provide accurate quantitative results, we performed additional experiments using high resolution imaging flow cytometry for pMLKL and cleaved caspase 8 (CC8) at two different MOIs and multiple time points (revised Fig. 3b,c and Supplementary Fig. 2). This assay was used to provide the quantitative results requested because its single cell resolution and debris filtering through image analysis gives improved quantification for epitope signal levels than does Western Blot analysis. The new summary data and analyses are presented in Fig. 3 (replacing the original Fig. 2F blot) and examples of the raw data underlying these results are shown in Supplementary Fig. 2. These experiments show a statistically significant increase in pMLKL and CC8 levels after NC/99 infection relative to Cal/09 infection.

<<Identifying necroptosis as the mode of cell death elicited by seasonal influenza infection relies on treatment of cells with drugs only, no genetic evidence is provided.>>

Authors' Response: To our knowledge, no lab has so far succeeded in CRISPR ablation of primary immune cells. We nevertheless attempted to ablate the expression of RIPK3 in human primary DC by transfecting riboproteins consisting of Cas9 nuclease coupled to the appropriate guide RNases. Unfortunately, we have been unable to achieve reliable and reproducible suppression without massive cell death resulting from this transfection. As an alternative and widely accepted validation approach, we have strengthened our pharmacological characterization to include confirmation of our results using multiple structurally distinct chemical inhibitors. We obtain consistent results using different inhibitors (Fig. 3a), which provides strong support for our hypothesis about the mechanisms of cell death.

<<How do the authors exclude that the partial protection from necroptosis that occurs following pandemic influenza infection is not due to reduced transfection with poly IC? What is the mechanism by which pandemic influenza-infected cells are protected from necroptosis? Note that in Figure 3A the amount of cell death inflicted by poly IC is significantly lower compared to panels b and c.>>

Authors' Response: The reviewer raises the question of whether the apparently protective effect of the pandemic virus is the result of reduced poly I:C transfection following pandemic influenza virus infection. To test this possibility, we quantified transfection of FITC-labeled poly I:C at the same time point studied (2 hours) after infection with either Cal/09, NC/99, or mock using imaging flow cytometry (revised Fig. 4e). Infection with the pandemic (Cal/09) or the seasonal (NC/99) IAV strain did not alter the levels of poly I:C transfection. This result excludes virus-specific differences in poly IC transfection levels as a contributing factor for the differences in necroptosis found.

We have determined that the HA segment from the pandemic virus suppresses RIPK3-mediated cell death. Further investigation of the underlying mechanisms through which HA protects these cells is discussed a subject for future study and various hypotheses are discussed in the manuscript. The reviewer notes differences in the absolute levels of cell death seen in different figure panels. These panels shown were generated using cells from different donors. While the relative levels of cell death with different conditions are found to be constant for different donors, the absolute percentage of cell death induced by any single intervention shows considerable variability from donor to donor. The variations in the percentage of cell death seen after poly I:C transfection when comparing cells obtained from different donors have no effect on the interpretation of the study. We have revised the legend of Fig. 4 to clarify this source of variation. Because of donor overall cell death variability, we replicate all experimental results using cells from at least three donors. The differences among the interventions (e.g. poly I:C vs. poly I:C + NSA) are comparably significant using cells from different donors. In addition, we have now extended the experiment originally shown in Fig. 3A (revised Fig. 4b) to evaluate the effect of different concentrations of poly I:C.

<<It is not clear what the assays of T cell proliferation are measuring. More detail is required to interpret these outcomes. Are influenza infected DC providing antigen for naive DC or eliciting naive DC cytokine production? [Note; is by crosspriming or direct priming?] What are the source of T cells and/or their specificity?>>

Authors' Response: We apologize for the lack of clarity in the original submission and appreciate the reviewer's request. We were interested in gaining insight into the possible immunological effects of this differential DC necroptosis induction by different viruses. Therefore, we studied the effects of DC necroptosis on DC capacity to activate T cells. Because these are human experiments, this cannot be directly studied using syngeneic T cells due to variability in memory T cells to different viruses, especially the recent Cal/09 virus. This and cross reactivity among different previous virus infections make syngeneic T cell study impractical. As an alternative, we studied allogeneic T cell activation, relying on the graft versus host difference to provide a general superantigen TCR activation. This thus provides the basis for comparing the levels of T cell proliferation induced by uninfected DC when exposed to DC infected with different viruses or DC treated with poly I:C. We show that DC exposure to necroptotic DC contributes significantly to the generation of T cell activation in this assay (revised Fig. 4f,g) and that DC exposure to DC infected with the wild-type and reassorted viruses containing the seasonal HA segment (that does not suppress necroptosis) leads to greater T cell activation. These data support a cross priming-like mechanism. Our results are consistent with the formulation that necroptotic DC release DAMPs that promote the maturation of DC and their activation of specific T cells (Fig. 6f). We have revised the manuscript to clarify the assay used and its rationale and interpretation.

<<HA is identified as the gene segment responsible for the enhanced death of DC infected with seasonal influenza. What is the mechanism responsible? Data in Figures 4d-f show a correlation with seasonal influenza virus infection or seasonal virus HA with the ability of infected DC to elicit increased T cell activation. Again, what is the mechanism responsible? There is no direct evidence linking these outcomes with necroptosis. Controls are required for the infectivity of the reassortment viruses.>>

Authors' Response: The Balachandran lab, our collaborators on this paper, as well as the Kanneganti lab, have recently implicated DAI/ZBP1 as the receptor linking IAV sensing to induction of necroptosis (Thapa *et al.*, 2016 Cell Host Microbe PMID 27746097, Kuriakose *et al.*, 2016, Sci Immunol PMID 27917412). This information has been added to the discussion. The specific HA sequences correlated with the suppression of necroptosis provide some clues about the underlying mechanism, which is discussed in the manuscript. We

show similar levels of infection with all constructs, excluding HA-mediated differences in infectivity as a mechanism. To further investigate the "HA effect", we have done additional experiments in which we exposed DC to either soluble pandemic HA or virus-like particles bearing the pandemic HA (see Supplementary Fig. 3). As both perturbations caused no inhibition of cell death, these results are consonant with the formulation that newly synthesized HA RNA or protein is necessary to inhibit necroptosis induction. We have expanded the discussion to propose potential mechanisms for HA-mediated necroptosis inhibition. We now include in the revised manuscript the infectivity data from the experiments illustrated in Fig. 5a,b, which show that the reassortment viruses have comparable infectivity (New Supplementary Fig. 6a,b). We have revised the results section as follows:

"It is noteworthy that all native and recombinant viruses were titrated at a comparable infectivity (with an MOI of 1), such that around 60% of DC expressed viral nucleoprotein NP (Supplementary Fig. 6a,b). Hence, the HA sequence-specific suppression of necroptosis is not the result of differences in infectivity."

<<It is not clear that monocyte-derived DC are a relevant cell type for infection with influenza virus infection. What is the evidence that these cells participate in the response to influenza virus infection in vivo? The manuscript would strongly benefit from data obtained in vivo. >>

Authors' Response: DC are responsible for virus recognition in the lung and subsequent initiation of the adaptive immune response. Several DC subtypes including inflammatory monocyte-derived DC are implicated in the staging of an anti-IAV immune response (for review, see Waithman and Mintern, 2012 Virulence PMID 23076333). While we have published *in vivo* mouse work on IAV-mediated necroptosis (Nogusa *et al.*, 2016 Cell Host Microbe PMID 27321907), mouse models have limitations in comparing the effects of different human IAV strains. In particular, we have previously reported that murine DC are resistant to infection by IAV and particularly resistant to pandemic IAV infection (Hartmann *et al.*, 2013 J.Virol. PMID 23192878). Therefore we cannot extend our studies to *in vivo* non human models. Instead, in order to look for *in vivo* relevance of our results, we compared the cell type composition of peripheral blood immune cells (inferred from global gene expression data) in patients who were infected with either a pandemic or a seasonal IAV strain. While there are caveats to this approach, discussed in the manuscript, this comparison revealed lower T cell proportions in patients infected with the pandemic IAV that is consonant with the *in vitro* effects we observed.

<<The Introduction & Discussion are significantly lacking in detail. The manuscript requires more detail regarding what is already known about (i) dendritic cells and influenza A virus infection, (ii) immune outcomes associated with pandemic versus seasonal influenza A virus strains and (iii) the role of necroptosis in influenza A virus. There are relevant publications for each of these topics, none of which are cited. >>

Authors' Response: The original manuscript was submitted as a short report, which severely limited the number of references cited and the length of the introduction and discussion. In this resubmission, we have substantially expanded the introduction and discussion, and include relevant citations pertaining to the areas raised by the reviewer.

Reviewer #2

<<1. The authors are using monocyte-derived DC in the experiments. There is no data presented showing the phenotype of the cells after the differentiation process. >>

Authors' Response: As requested, we include flow cytometry phenotyping data on the monocytes before culture and after 5 days of culture with GM-CSF and IL-4, which transforms them into the mDC used for this study (Supplementary Fig. 1).

<<2. The authors present data showing the DC death after seasonal IAV infection was via necroptosis and RIPK3-dependent. This data generally replicates the data recently published by Nogusa *et al.* in *Cell Host & Microbe*. Interestingly, the authors of this manuscript were co-authors on the Nogusa *et al.* publication, which went into more extensive detail describing the mechanism of the IAV-induced necroptosis. Granted, the Nogusa *et al.* paper did everything in mouse cells and used the common mouse-adapted IAV strains PR/8 and x-31 - while the submission by Hartmann *et al.* uses human DC and human IAV strains. Regardless, the overall novelty of the manuscript under review is significantly decreased because of this other paper already published. >>

Authors' Response: The overall goal of our study was not to investigate IAV-induced necroptosis (the subject of Nogusa *et al.*, 2016 *Cell Host Microbe* PMID 27321907), but rather to compare the effects of the wild-type human pandemic and seasonal strains in cells of their NATURAL host, the human. Increasing evidence suggests that cell death outcomes can be very different in cells of natural host species, versus species-mismatched cells. For example, HSV-1 activates necroptosis in murine cells, but blocks necroptosis in human cells (for review: Guo *et al.*, *Med Microbiol Immunol.* 2015 PMID: 25828583 as well as Huang *et al.*, 2015 *Cell Host Microbe* PMID; 25674982 and Guo *et al.*, 2015 *Cell Host Microbe* PMID; 25674983)

The research leading to the present manuscript was motivated by our previous report (Hartmann *et al.*, 2015 *J. Virol.* PMID 26223639), which revealed dramatic differences in the global gene response pattern elicited by infection with pandemic or seasonal human IAV in human DC. While the present study and Nogusa *et al.* both involve aspects of IAV-mediated necroptosis, they are entirely distinct and complementary in focus, impact and novelty. Our study (i) reveals a difference in necroptosis induction between pandemic and seasonal IAV, (ii) shows an actual inhibition of necroptosis induction by pandemic IAV – which appears to represent a novel immune antagonistic strategy, (iii) identifies the pandemic HA viral segment as the determinant of necroptosis inhibition, (iv) links necroptosis inhibition to an overall lower T cell response *in vitro* as well as *in vivo*. These four points represent an important breakthrough in understanding host specific mechanisms developed by pandemic IAV.

<<3. The T cell proliferation data in Figure 3d does not make sense. First, what is driving the T cell proliferation? There is no Ag provided. Second, while the authors include necrosulfonamide to block necroptosis, the inclusion of cells killed by some other mechanism (e.g., apoptosis) would have been a nice control. Third, the authors do not provide any data examining the phenotype of the uninfected bystander DC presumably being activated by DAMPs. >>

Authors' Response: Reviewer Point 1) We repeat here the response from above to a similar comment by reviewer 1. We apologize for the lack of clarity in the original submission and appreciate the reviewer's request. We were interested in gaining insight into the possible immunological effects of this differential DC necroptosis induction by different viruses. Therefore, we studied the effects of DC necroptosis on DC capacity to activate T cells. Because these are human experiments, this cannot be directly studied using syngeneic T cells due to variability in memory T cells to different viruses, especially the recent Cal/09 virus. This and cross reactivity among different previous virus infections make syngeneic T cell study impractical. As an alternative, we studied allogeneic T cell activation, relying on the graft versus host difference to provide a general

superantigen TCR activation. This thus provides the basis for comparing the levels of T cell proliferation induced by uninfected DC when exposed to DC infected with different viruses or DC treated with poly I:C. We show that DC exposure to necroptotic DC contributes significantly to the generation of T cell activation in this assay (revised Fig. 4f,g) and that DC exposure to DC infected with the wild-type and reassorted viruses containing the seasonal HA segment (that does not suppress necroptosis) leads to greater T cell activation. These data support a cross priming-like mechanism. Our results are consistent with the formulation that necroptotic DC release DAMPs that promote the maturation of DC and their activation of specific T cells (Fig. 6f). We have revised the manuscript to clarify the assay used and its rationale and interpretation.

Reviewer Point 2) The goal of the paper was not to characterize the well-established potential for necroptosis to release DAMPs and to contrast this with apoptosis, but to compare the effects of infection with pandemic and seasonal IAV. We find that pandemic IAV suppresses cell death and suppresses *in vitro* T cell activation, in dramatic contrast to the effects of seasonal IAV infection. While comparing the effects of necroptotic and apoptotic DC on inducing other DC to cause T cell activation is an interesting question, it does not appear relevant to the present study. Nonetheless, to address this issue, we now show that exposure of uninfected DC to DC in which apoptosis is induced by staurosporine (or DC infected by the pandemic Ca) I/09 virus do not stimulate T cell proliferation. This is in contrast with DC exposed to necroptotic DC generated by infection with seasonal NC/99 virus or transfected with Poly I:C, which stimulate T cell proliferation. (Supplementary Fig. 5)

Reviewer Point 3) We have phenotyped these cells by analyzing maturation marker expression in DC that were either in direct contact with infected/necroptotic DC, or were exposed to paracrine signaling from infected/necroptotic DC via a transwell system (Supplementary Fig. 4).

<<4. *The data in Figure 4 is the most interesting, which identified HA as the viral component that induced necroptosis. The authors need to expand their investigation of the differences in the HA proteins from the different IAV used. At least, a better discussion of potential mechanisms defining the "HA effect" should have been done. >>*

Authors' Response: To further investigate the "HA effect", we have done additional experiments in which we exposed DC to either soluble pandemic HA, or virus-like particles bearing the pandemic HA (see Supplementary Fig. 3). As both perturbations caused no cell death inhibition, these results are consonant with the formulation that newly synthesized HA RNA or protein is necessary to inhibit necroptosis induction. We have expanded the discussion to propose potential mechanisms for HA-mediated necroptosis inhibition, including its possible effects on activation of the newly-discovered sensor of IAV-RIPK3 cell death, DAI (Thapa *et al.*, 2016 Cell Host Microbe PMID 27746097, Kuriakose *et al.*, 2016, Sci Immunol PMID 27917412).

<<5. *The Discussion is almost non-existent, and the text that is there is mostly a rehashing of the results section. >>*

Authors' Response: The original submission was a short report format. We have now considerably expanded the discussion.

Reviewer #3

<<1) *The authors largely present their findings in a clear and succinct manner, however the introduction and discussion are too short. These sections need to be expanded. The introduction does not provide a sufficient*

review of the existing literature and needs to be developed into a coherent form that provides the reader a strong basis for comprehending the manuscript going forward. For instance what is the current understanding of the role of cell death in virus (IAV)-host interactions (e.g. how do viruses like IAV manipulate cell death to perhaps promote infectivity and/or how does the host exploit cell death to its benefit)? What is our current understanding of the arms race between IAV and the host? Once a sound review is provided, the authors need to build on it to pose the question under investigation in the current manuscript. As the intro currently stands, it is left to the reader to deduce or else to extensively inspect the literature to understand a) what the question/hypothesis under investigation is, b) why it is novel and c) why was it posed in the light of current literature? Similarly, the discussion needs to integrate the authors' findings with current cell death / IAV literature and perhaps speculate on its importance, rather than serve as a short summary of the results. >>

Authors' Response: The original manuscript was submitted as a short report, which severely limited the number of references cited and the length of the introduction and discussion. In this resubmission, we have substantially expanded the introduction and discussion, and include relevant citations pertaining to the areas raised by the reviewer.

<<2) Fig. 2d: Cell death in necrosulfonamide (MLKL inhibitor) or necrosulfonamide+ZVAD treated cells is approx. 50% lower than that in untreated or ZVAD alone conditions at baseline (i.e. in the absence of NC/99 virus infection). In the virus-infected condition, one sees the same 50% reduction in cell death between these treatments. Therefore the effective fold increase in cell death upon virus infection is ~2 fold irrespective of MLKL inhibition. From these data, one cannot convincingly conclude that the reduction seen with MLKL inhibitor is due to an active role for MLKL during NC/99 infection, and not a result of reduced responsiveness due to baseline status of cells treated with MLKL inhibitor. The effects of necrosulfonamide in Fig. 2e are more convincing, but it is unclear which panel represents real biology (2d or 2e). Please clarify and replace the data in Fig. 2d with an experiment devoid of significant baseline effects of the inhibitor alone. >>

Authors' Response: We used additional, structurally distinct, chemical inhibitors and also repeated the whole experiment using all the relevant inhibitors simultaneously. Thus, we combined all the panels illustrating the use of inhibitors into a single panel (Fig. 3a). None of the inhibitors had a significant effect in mock-infected DC. The new figure now convincingly demonstrates that IAV activates both RIPK3-driven necroptosis and apoptosis in infected cells, and that the combination of a RIPK3 inhibitor and zVAD are needed for full rescue of viability. Interestingly, NSA blocks death on its own, suggesting that RIPK3 activity is necessary both for activation of MLKL and necroptosis and for induction of caspase activity and apoptosis.

<<3) The authors need to provide mechanistic insight into how/why DC necroptosis enhances T cell proliferation/adaptive immunity. Figure 3d contains some good data. The authors should determine how incubating DCs with necroptotic cells influences a) their ability to make cytokines, b) express co-stimulatory molecules and c) to present antigen, and how this affects the ability of T cells to make cytokines. >>

Authors' Response: To provide more mechanistic insight into how DC necroptosis enhances T cell activation, we examined maturation marker expression in naïve DC exposed to infected/necroptotic DC, either through direct contact or through a transwell system (Supplementary Fig. 4). Using allospecific stimulation of T cells by DC as our experimental approach, we co-cultured infected DC with T cells. Our results show that necroptotic DC can influence uninfected DC via direct contact to more efficiently induce T cell activation. We have clarified this in the results, methods, and discussion.

<<4) It is well known that small molecule inhibitors can have off target effects. Therefore, data from cells where RIPK3, MLKL and RIPK1 expression has been ablated with either siRNA or CRISPR/Cas9 are absolutely necessary to complement the inhibitor data (Figs. 1 and 2). Where available, additional inhibitors of RIPK3 and MLKL should be used. >>

Authors' Response: In the revision, we have tested additional structurally distinct chemical inhibitors for RIPK3 and MLKL (see Fig. 3a) to minimize possible off-target effects. We have been unable to obtain an effective level of genetic ablation by CRISPR in primary human immune cells while maintaining cell viability. To our knowledge, CRISPR has not been carried out successfully in primary human immune cells. The GSK inhibitors utilized are highly specific. Furthermore, our collaborators (Nogusa *et al.* 2016 Cell Host Microbe PMID: 27321907) have demonstrated in mouse knockout studies that necroptosis induction by IAV is mediated by RIPK3-MLKL, whereas apoptosis is mediated by RIPK3-FADD/caspase8. Our chemical inhibitor data are supported by these mouse data, strongly suggesting that IAV induction of cell death is conserved between humans and mice.

<<5) HA is tightly linked to influenza entry into cells. Therefore, the data shown in Figure 4 could arise from differences in the infectivity of the recombinant viruses towards DCs. Demonstrating equal infectivity and viral replication by assaying for expression of NP, NS1 or another non-structural protein is absolutely essential for these data to be indicative of a role for HA in preventing DC necroptosis. Given that this is the main point of the paper, the authors should show that HA alone inhibits necroptosis. This can be done by transfecting purified HA or plasmids expressing HA from seasonal or pandemic IAVs into DCs and examining poly(I:C)-induced necroptosis in these cells. >>

Authors' Response: We now include in the revised manuscript the requested infectivity data from the experiments illustrated in Fig. 5a,b, which show that the reassortment viruses have comparable infectivity (New Supplementary Fig. 6a,b). Thus, we exclude HA-mediated differences in infectivity as a mechanism. We have revised the results section as follows: "It is noteworthy that all native and recombinant viruses were titrated at a comparable infectivity (with an MOI of 1), such that around 60% of DC expressed viral nucleoprotein NP (Supplementary Fig. 6a,b). Hence, the HA sequence-specific suppression of necroptosis is not the result of differences in infectivity." To further investigate the "HA effect", we have done additional experiments in which we exposed DC to either soluble pandemic HA or virus-like particles bearing the pandemic HA (see Supplementary Fig. 3). Both perturbations caused no cell death inhibition. Thus, together with the data presented in Fig. 2, these results are consonant with the formulation that newly synthesized HA RNA or protein is necessary to inhibit necroptosis induction. We have expanded the discussion to propose potential mechanisms for HA-mediated necroptosis inhibition.

<<6) Extended Data Fig. 3: Here one hardly sees any T cell proliferation with NC/99 infected DCs compared to mock infected DCs which is contrary to that shown in Fig. 4e. This figure needs to be changed/redone. Please also add representative CFSE FACS plots to Fig. 4e and Extended Data Fig. 3. >>

Authors' Response: The percentage of T cell proliferation induced in the presence of mock-infected DC (in Fig. 4e - now Fig. 6a) differs from that induced in the presence of mock-infected DC, which were then transfected with poly I:C (in Extended Data Fig. 3 - now Fig. 6c), as poly I:C induces necroptosis. Therefore, these two controls cannot be compared to each other. To improve clarity, we have added "Poly I:C" in the bar description in Fig. 6c. Representative flow cytometry plots showing the dilution of CFSE have been added for both experiments (Fig. 6b,d).

<<7) Fig. 4f: The seasonal influenza (Bri/07) data are only from 9 individuals and in 3 individuals (i.e. a third of the sample size) the CD8 T cell proportion totally crashes compared to the pre Bri/07 infection. Given the heterogeneity in the post Bri/07 condition, this is a sample size too small to draw any biologically meaningful conclusions. To be suitable for publication, the authors need to reproduce this trend by pooling data from other seasonal IAV studies to get a larger sample size. >>

Authors' Response: The results presented achieve statistical significance. We revise the manuscript to include the caveat that this conclusion, while significant, is based on a small number of infected individuals. Combining data across studies requires very large numbers due to confounding batch effects that are present in genome scale assays. We have not been able to locate or obtain any larger controlled study of gene responses during seasonal IAV infection.

<<8) Why does CD8 T cell abundance in Cal/09 infected individuals fall below that in the pre infection condition? This seems unusual but the authors don't comment on it. Please explain. >>

Authors' Response: The patient samples were blood draws obtained after onset of symptoms. The peripheral blood T cell proportion is dependent on the net T cell flux (the rate of T cell entry into blood minus the rate of T cell exit from blood). T cell entry should be increased when T cell proliferation occurs. The results are consistent with the formulation that the lower T cell proportion in Cal/09-infected patients is the reduction of T cells due to infected tissue homing and T cell efflux from blood that is not offset by an increase in T cell proliferation (presumably due to the suppression of necroptosis by the pandemic virus). We have clarified this in the discussion.

Minor comments:

<<1) How was the plot in Fig. 1a generated? What is the intact nucleus score and how was it computed? Y-axis labels should be provided and the basis for assigning a specific score to a cell should be described in the methods or in the figure legend. >>

Authors' Response: We provide a detailed explanation for the computation of the intact nucleus score in the methods section.

<<2) Insets in Figs. 1b and 2c should have their own designation within Figs. 1 and 2. >>

Authors' Response: Infectivity data in Fig. 1 are now shown in the separate panels 1c and 1f.

<<3) Concentrations of actinomycin D, all other inhibitors and poly I:C used in each instance should be specified in the methods section and figure legends. >>

Authors' Response: Concentrations of all chemicals are now listed in the methods as well as in the figure legends.

<<4) *Figure 2f and Page 5 (Section titled "NC/99-induced cell death is RIPK3-dependent"):* The authors' claim that infection with the pandemic Cal/09 IAV strain did not significantly increase MLKL phosphorylation is contrary to what their western blot data in Fig. 2f show. The western blot clearly shows that following Cal/09 IAV infection there is an increase in MLKL phosphorylation at 8 hours compared to mock infection, albeit lesser than that seen with the NC/99 strain. The statement on Page 5 should therefore be reworded to accurately reflect this data. >>

Authors' Response: We reanalyzed the phosphorylation of MLKL and cleavage of caspase 8 by flow cytometry at 2 different MOIs. Results show that Cal/09 does not induce phosphorylation of MLKL. This single cell-based assay is more definitive than Western blot analysis, where many factors (for example the fraction of surviving cells) can influence the readout.

<<5) *Fig. 4d: How was the phylogenetic analysis done? Please provide details in the methods section.* >>

Authors' Response: A description of the phylogenetic analysis has been added in the methods section.

<<6) *Fig. 3c: How was this experiment done? Were cells pretreated with different MOIs of Cal/09 followed by Poly I:C transfection? Please fix the figure legend for this panel.* >>

Authors' Response: We have revised the legend of Fig. 3c (now Fig. 4d) as requested.

<<7) *Extended Data Fig. 4 is not cited anywhere in the main text except the methods. Because this figure is lays the foundation for the subsequent CD8 T cell analysis please discuss it in the main text.* >>

Authors' Response: Extended Data Fig. 4 (now Supplementary Fig. 6c,d) supports the validity of the post-hoc cell type proportion analysis using gene expression data. We have referred to it in the results.

<<8) *Extended Figure 2a: Increase in MFI of pMLKL upon NC/99 infection is very modest, which isn't completely consistent with the much larger increase in pMLKL shown in Fig. 2f. Please explain. Please also show representative FACS plots for pMLKL* >>

Authors' Response: We reanalyzed the phosphorylation of MLKL and cleavage of caspase 8 by high resolution imaging flow cytometry at 2 different MOI (Fig. 3b,c). Results show that Cal/09 does not induce phosphorylation of MLKL. Additionally, raw imaging flow cytometry data are now provided in Supplementary Fig. 2.

<<9) *Extended Figure 2b: What is the scale on the X-axis? A graph corresponding the flow plots (as in Extended Fig. 2a) should be shown. Please also state the number of replicates and repeats for data, and what the error bars represent in Extended Figs. 2 and 3.* >>

Authors' Response: We have re-annotated all the figures and indicated the number of experiment replicates and biological sample replicates in the figure legends.

<<10) *Extended Figure 2c: Is the X-axis label correct? A concentration of mg/uL of Poly I:C sounds quite high.*
>>

Authors' Response: Thank you for noting this font format error. The units should be [µg/mL]. This is now corrected in Fig.4a.

<<11) *When processing data for CellCODE analysis, can the authors explain why they select only the probe with the highest average expression for each gene (as mentioned in the methods section on Page 12)? Part of the advantage of having multiple probesets for each gene on microarray is to bypass any artifacts that may arise due to single probesets.* >>

Authors' Response: We agree with the reviewer that, in principle, multiple probe sets for each gene on a microarray are partly designed to bypass any artifacts that may arise due to single probe sets. Other reasons include cases of alternative splicing or use of alternative poly(A) sites, complicating any genome-wide schemes that would derive gene expression levels from multiple probe sets (for reference, see <https://www.ncbi.nlm.nih.gov/pmc/articles/PMC1784106/>). Thus, in practice, single probe sets are often used, and probe selection is based on one of many reasonable choices, such as the median expressing probe or the choice we made, the probe with the highest average expression across the time-course for each gene. This scheme successfully selected the most robustly expressed probe set (isoform).

<<12) *Page 8, Discussion: Please fix the following sentence "As transfection of the viral RNA mimetic Poly I:C by alone induces necroptosis (Extended Data Fig. 2c), the inhibition"*. Delete 'by'. >>

Authors' Response: This sentence has been deleted in the revised and expanded discussion.

Reviewers' comments:

Reviewer #1 (Remarks to the Author):

While the manuscript is improved, I remain unconvinced of the significance of the overall findings given the previous literature. Data regarding seasonal influenza infection eliciting uninfected and infected DC interactions, together with the role of HA in suppressing necroptosis are interesting but currently these are not sufficiently developed to raise the significance of the overall findings.

Minor comments.

1. Introduction. "and/or direct contact with infected DC initiate the maturation of uninfected DC" Could the authors provide a citation for this statement?
2. Results text for Figure 1 should describe which strains are seasonal and which are pandemic.
3. Figure 3b requires statistics to allow conclusions to be drawn. The x-axis requires units of time.
4. Figure 4e should be represented as a graph of MFI, with statistics included.
5. Data with DC maturation remains weak or inconclusive, this could be improved to elevate the manuscript's significance.

Reviewer #2 (Remarks to the Author):

Hartmann et al. have submitted a revised manuscript examining the differences between seasonal and pandemic IAV in their ability to induce DC death. The authors show the seasonal IAV caused DC death, while the pandemic strains did not. Data is presented to suggest the death occurs by both apoptotic and necroptotic means. The key finding is that the HA genomic segment is what regulates the DC cell death. The authors used various recombinant IAV where the HA segments were swapped between seasonal and pandemic IAV to reach this conclusion.

This revised manuscript is improved over the original submission, but there are still a few weaknesses the authors need to address:

1. In Figure 1, the quantification of cell death by nuclear fragmentation and cell morphology is not convincing. It would seem this approach overestimates the amount of cell death, as the frequencies of dead cells are much higher in Figure 1A vs. 1G. There was no discussion why the TUNEL and LDH release numbers were so low in comparison.
2. In Figure 2, the authors conclude that the extrinsic cell death pathway was not engaged in the DC cell death. None of the experiments presented can determine this fact. For example, Fas-induced death in activated T cells can occur in a cell autonomous manner - but it occurs thru the interaction between Fas and FasL.

The authors did not include any inhibitors of the known extrinsic pathways in the experiments. What happens when TNFR:Fc, Fas:Fc, or TRAIL-R:Fc are added to the cultures? Moreover, what is the expression of TNF, FasL, and TRAIL on the DC after infection? For example, IFN-stimulated and virally-infected DC can express TRAIL. Any of these

It is unclear how the authors reached the conclusion that the DC cell death in the mixed cultures was cell intrinsic.

The authors' conclusions cannot be supported based on the data currently presented in Figure 2.

3. It is difficult to see any differences in the overlaid histograms in Fig 3 (as well as the rest of the figures).

Reviewer #3 (Remarks to the Author):

Hartmann et al present a revised version of their study showing a difference in the ability of pandemic vs seasonal IAV strains to induce DC necroptosis, and claim that the ability of pandemic strains to inhibit necroptosis is dependent on the HA component. While the new version expands on the introduction / discussion sections to include relevant literature and some new data alleviate some initial concerns about the lack of controls (most importantly controls for the infectivity of reassortment viruses, Fig. S6A-B), other new pieces of data raise substantial questions about what is going on in this system. Data trying to mechanistically link necroptosis to T cell activation, are inadequate and unconvincing, in many cases conclusions drawn are confusing and inconsistent with the actual data presented. The manuscript is a difficult read because of the way data are presented in places and the way it is written. Overall the advance over the previous version is incremental, in some cases confusing in the light of new information provided. As it currently stands, substantial development of the manuscript, its presentation and data would be required for this manuscript to be an important advance or influence thinking in the field. The authors would need to show how pandemic HA exerts its inhibitory effect and further develop the effects on T cell activation.

1. Lentiviruses and retroviruses are a requisite vehicle to deliver probes for genetic ablation into primary cells, at the least siRNA or shRNA strategies should be employed to complement the RIP3, MLKL and RIP1 inhibitor data.

2. Fig. 3A: Why do MLKL inhibitors (NSA and SYN-1215) inhibit cell death in response to NC/99 infection in absence of z-VAD? Presumably here cells are dying by apoptosis. RIPK3 has been implicated in both apoptosis (kinase activity independent) and necroptosis (kinase activity dependent), however the role of MLKL in apoptosis is not known. Overall it is unclear what is happening here with the MLKL inhibitors making another means of MLKL ablation all the more important. If MLKL silencing / genetic ablation shows a similar effect, the authors need to comment on this.

3. Fig.3C, 4G, 6B, 6D, S4: throughout the paper FACS histograms are shown / overlaid (in solid filled colors) in a way that makes it very difficult if not impossible to see what the distributions for each condition look like. Many histograms are hidden behind others, in figure S4 representative dot plots with gating strategy on NP negative cells are not shown. In general, the manner in which these data are presented is not amenable to technical or scientific review.

4. All of the data in the paper show that cell death in DCs in the absence of Z-VAD (presumably apoptosis) is as efficient and in some cases more efficient than that in the presence of Z-VAD (necroptosis) (Fig 3A and 4B). Figs 3A and 4B are the only figures where the contribution of necroptosis assessed by caspase inhibition with Z-VAD is shown. In all other instances where the inhibitory effect of Cal/09 or the potentiating effect of NC/99 on cell death and T cell proliferation is evaluated (Figs. 4-6 and elsewhere), treatments with viruses and/or poly I:C are in the absence of Z-VAD. Without caspase inhibition in these experiments, how do the authors know that cells are undergoing immunogenic / necroptotic death as opposed to non-immunogenic apoptotic death? Granted that seasonal strains and poly I:C can induce necroptosis (3A, 4B), but how does one know that seasonal strains are indeed inducing necroptosis and pandemic strains are inhibiting necroptosis (and not another form of death) in these experiments? This is one of the primary claims of the study and there are no solid data in the paper to support this main conclusion. Fig. 3C implies that NC/99 is more efficient than Cal/09 at inducing both MLKL phosphorylation (necroptosis) and caspase 8 cleavage (apoptosis), and Fig. 3A shows that MLKL inhibitors can inhibit apoptosis. Taken together the data are thoroughly confusing. It is obscure how the authors can draw any sound conclusions about modulation of necroptosis by seasonal and pandemic IAV from these data.

5. Figure S4: Conclusions are at odds with the data. Text on Pg. 6, line 174 reads: "Comparison of the effects of direct cell contact of naïve DC and infected DC (mixed DC co-culture) to that of only factors secreted by infected DC (transwell DC culture) shows that after seasonal IAV infection, direct contact leads to an increase in CD86, CCR7 and HLA-ABC levels in the naïve DC (Supplementary Fig. 4). In contrast, the effects of exposure to DC infected with the pandemic Cal/09 infection, which blocks DC necroptosis, are restricted to only an increase in CCR7 levels in naïve DC; this increase was not enhanced by direct contact. No increases in the other maturation markers assayed in naïve DC were seen after direct or transwell exposure to pandemic virus-infected DC". Upregulation of markers except CCR7 in NC/99 mixed DC co-cultures is at best modest. Moreover, the flow plots as presented, show that direct contact of pandemic Cal/09 infected DCs with naïve DCs upregulates not only CCR7 but also CD86 and HLA-ABC in naïve DCs. Therefore, the authors conclusions are not supported by their data.

6. Pg 21, Figure 4g legend. last line reads: "g shows representative imaging flow cytometry plots". Is this correct or are the CFSE dilution plots regular flow plots?

7. Figure 6: The conclusion that pandemic IAV causes reduced T cell proliferation is not supported by the data. In Fig. 6E, although more number of T cells incubated with naïve

DCs crossprimed by NC/99 infected DCs enter division compared to T cells incubated with naïve DCs crossprimed by Cal/09 infected DCs, the latter divide more. The data as presented show that T cells incubated with DCs crossprimed by pandemic Cal/09 infected DCs that do enter division are more efficient proliferators compared to their NC/99 counterparts.

8. Figure 6C-D: It is unclear what the authors are trying to accomplish here. Where does the inhibitory effect of Ca/09 on T cell proliferation come from? The claim that DC markers are affected upon direct contact with DCs infected with NC/99 vs Cal/09 (Fig. S4 discussed previously) is unconvincing. The ability of DCs to make cytokines, which may provide insight into the mechanism of Cal/09 inhibition, is not evaluated.

9. Figure 6E: These data remain unconvincing. I agree that they may reach statistical significance, but the heterogeneity in the post Bri/07 condition where CD8 counts in 3 out of 9 individuals completely crash make it very difficult to believe that the data are biologically significant. If the authors' in vitro data is to any degree reflective of the in vivo situation in humans, one would expect to see greater T cell proliferation aka greater numbers of circulating T cells in the post-BRI condition, which is not the case. This doesn't mean that the in vitro data is not useful, just that the human data is not at a point to draw any conclusions that support the authors' other data.

10. Supplementary fig. 1 is not cited in the text, only in the methods section.

11. Supplementary Fig. 2 legend has wrong text. Delete: "MFI, mean fluorescence intensity. Shown are representative imaging flow cytometry plots from a replicate at 6 h post-infection or treatment." In this figure, representative cells, not plots, are shown.

12. Pg. 23, line 786: Supplementary Figure 6 should be Supplementary Figure 5. Pg. 24, line 794: Supplementary Figure 5 should be Supplementary Figure 6. Line 796-797: Delete "c Comparison of T cell proliferation by necroptotic and apoptotic cells."

Response to Reviewers

Note: Reviewers' comments are in <<italics>> and authors' responses are in plain text.

Authors' Response Introduction:

We have performed additional experiments and revised the paper to address each of the reviewers' comments. Reviewers' comments are quoted below in italics, while our responses follow each comment. In short, we completed genetic ablation experiments in human primary immune cells, performed several requested control experiments, and substantially improved the study addressing the upregulation of maturation markers in uninfected DC in direct contact with infected necroptotic DC. We have made additional revisions to address all of the reviewers' concerns.

Reviewer I:

<<Introduction. "and/or direct contact with infected DC initiate the maturation of uninfected DC" Could the authors provide a citation for this statement?>>

Authors' Response: We added Borderia et al. (PMID: 18981106) as a citation, as this study showed that paracrine signaling from infected DC can alert uninfected DC to mature into an antiviral activated state.

<<Results text for Figure 1 should describe which strains are seasonal and which are pandemic.>>

Authors' Response: We added the abbreviations for the two seasonal strains (NC/99 and Tx91) as well as the abbreviations for the two pandemic strains (1918, Cal/09) in the Results section referring to Fig 1. For additional clarity, we now identify the seasonal and pandemic strains in the legend of Fig. 1.

<<Figure 3b requires statistics to allow conclusions to be drawn. The x-axis requires units of time>>

Authors' Response: ANOVA analysis shows that NC/99 and the positive control TCZ (in the case of pMLKL) are associated with a significant increase in MLKL phosphorylation and caspase 8 cleavage when compared to infection by Cal/09 (New revised Fig. 3c). To make the data easier to interpret for the reader, we also plotted only the most informative 5-hour post infection time point in the revised figure.

<<Data with DC maturation remains weak or inconclusive, this could be improved to elevate the manuscript's significance.>>

Authors' Response: By extending the time of the co-culture of infected DC with naïve DC to 18 hours, we were able to confirm significant upregulation of the maturation markers HLA-DR, HLA-ABC, CD40, CD86, and CCR7 in the naïve DC that were in direct contact with NC/99-infected DC. These data are shown in the new Fig. 5. Consonant with previous data, direct contact between DC and T cells (in comparison with secreted factors alone via a transwell system) is required for activation by NC/99-infected immunogenic DC.

Reviewer #2

<<1. In Figure 1, the quantification of cell death by nuclear fragmentation and cell morphology is not convincing. It would seem this approach overestimates the amount of cell death, as the frequencies of dead cells are much higher in Figure 1A vs. 1G. There was no discussion why the TUNEL and LDH release numbers were so low in comparison. >>

Authors' Response: While the large difference between the effects of seasonal vs. pandemic strains was observed reproducibly in DC derived from several dozens of different donors, the absolute percentage of dead cells after mock treatment and after infection with a seasonal virus varies in DC obtained from different donors. We obtain comparable cell death results using Tunel and imaging flow cytometry when analyzing cells obtained from the same donor. However, in Fig. 1 we cannot show results from the same donor in panels 1a and 1g as the Tunel and infectivity assays cannot be done using the BSL3 Brevig/1918 virus strain. In addition, the LDH release assay is normalized relative to total cell lysis, and provides cell death measurements that are on a different scale than Tunel and imaging flow. Nevertheless, it is important to note that using LDH release assays, a significant difference in the percentage of cell death caused by seasonal vs. pandemic IAV could also be confirmed. To avoid confusion due to the variation in overall cell death from donor to donor following IAV infection, we now emphasize this variation in the legend and have selected a donor for the revised Fig. 1g that has a similar level of cell death to that obtained in Fig. 1a. We also revised the legend to indicate which panels come from cells obtained from different donors.

<<2. In Figure 2, the authors conclude that the extrinsic cell death pathway was not engaged in the DC cell death. None of the experiments presented can determine this fact. For example, Fas-induced death in activated T cells can occur in a cell autonomous manner - but it occurs thru the interaction between Fas and FasL. The authors did not include any inhibitors of the known extrinsic pathways in the experiments. What happens when TNFR:Fc, Fas:Fc, or TRAIL-R:Fc are added to the cultures? Moreover, what is the expression of TNF, FasL, and TRAIL on the DC after infection? For example, IFN-stimulated and virally-infected DC can express TRAIL. Any of these it is unclear how the authors reached the conclusion that the DC cell death in the mixed cultures was cell intrinsic. The authors' conclusions cannot be supported based on the data currently presented in Figure 2.>>

Authors' Response: To address the reviewer's concerns about extrinsic cell death induction, we measured the expression of a panel of cytokines in DC infected with either pandemic Cal/09, seasonal NC/99, or seasonal Tx/91 (revised Fig. 2a). As requested, we measured cell death induction by NC/99 in the presence of antibodies against FASL or TRAIL, or a TNF α antagonist; these new data are presented in the revised Supplementary Fig. 2. TNF α was relatively poorly induced by IAV infection, and the level of induction tended to be higher following pandemic IAV infection than following cell death-inducing seasonal IAV infection (Fig. 2a). Furthermore, TNF α antagonist showed no inhibitory effect on seasonal NC/99-induced cell death (Supplementary Fig. 2a) and co-culture in a transwell system did not induce cell death (Fig. 2b). These data support the conclusion that TNF α signaling does not play a detectable role in cell death induction by seasonal IAV infection. Neither addition of anti-TRAIL nor anti-FASL antibody affected DC cell death induction by NC/99. Furthermore, TRAIL could not be detected above the assay detection limit after infection with any of the IAV strains (Fig. 2a). Also, exposing DC to high doses of FASL did not induce cell death (data not shown), making FAS-FASL-mediated cell death in our system very unlikely, which is in concordance with a report by Rescigno

et al. (PMID: 11104808). Overall, these data support the conclusion that external signaling is not responsible for the induction of cell death by seasonal IAV infection.

<<3. *It is difficult to see any differences in the overlaid histograms in Fig 3 (as well as the rest of the figures).>>*

Authors' Response: We have revised the presentation of the flow cytometry data to make the comparison between samples easier to observe.

Reviewer #3

<<1. *Lentiviruses and retroviruses are a requisite vehicle to deliver probes for genetic ablation into primary cells, at the least siRNA or shRNA strategies should be employed to complement the RIP3, MLKL and RIP1 inhibitor data.>>*

Authors' Response: In our hands, siRNA delivery causes high cell mortality and poor target suppression in monocyte-derived DC. After extensive trials, we have succeeded in genetically ablating RIPK1, RIPK3, FADD, MLKL, and DAI using CRISPR ribonucleoproteins delivered via electroporation. To our knowledge, this is the first CRISPR genetic ablation study performed in this primary DC model. In these new experiments, we observed that ablation of RIPK3 and combinatorial ablation of MLKL and FADD but not MLKL alone could significantly reduce cell death induction by NC/99 infection. These results are consonant with the observations of Nogusa et al. (PMID: 27321907) in mouse. Furthermore, by knocking down DAI we obtained results in human cells that are consonant with previous observations in mouse by Thapa et al. (PMID: 27746097) that DAI acts as a viral RNA sensor that induces RIPK3-dependent cell death.

<< 2. *Fig. 3A: Why do MLKL inhibitors (NSA and SYN-1215) inhibit cell death in response to NC/99 infection in absence of z-VAD? Presumably here cells are dying by apoptosis. RIPK3 has been implicated in both apoptosis (kinase activity independent) and necroptosis (kinase activity dependent), however the role of MLKL in apoptosis is not known. Overall it is unclear what is happening here with the MLKL inhibitors making another means of MLKL ablation all the more important. If MLKL silencing / genetic ablation shows a similar effect, the authors need to comment on this. >>*

Authors' Response: The genetic ablation experiments showed that knockdown of RIPK3 could significantly reduce cell death induction. Knockdown of MLKL without simultaneously knocking down FADD did not affect seasonal virus-induced cell death. These data suggest that, unlike the more specific genetic ablation of MLKL, the chemical MLKL inhibitor NSA also interferes with RIPK3-mediated caspase activation in this experimental system. The text has been revised to clarify this point.

<< *Fig.3C, 4G, 6B, 6D, S4: throughout the paper FACS histograms are shown / overlaid (in solid filled colors) in a way that makes it very difficult if not impossible to see what the distributions for each condition look like. Many histograms are hidden behind others, in figure S4 representative dot plots with gating strategy on NP negative cells are not shown. In general, the manner in which these data are presented is not amenable to technical or scientific review. >>*

Authors' Response: We have revised the presentation of the flow cytometry data to make the comparison between samples more apparent. We added a gating strategy figure (new Supplementary Fig. 7) and the data from the previous Supplementary Fig. 4 have been moved to the revised Fig. 5. We also now include a

summary plot with statistical analysis of the repeated experiments. We appreciate the reviewer's comments and think the revisions have made the data and their interpretation more accessible to the reader.

<< 4. All of the data in the paper show that cell death in DCs in the absence of Z-VAD (presumably apoptosis) is as efficient and in some cases more efficient than that in the presence of Z-VAD (necroptosis) (Fig 3A and 4B). Figs 3A and 4B are the only figures where the contribution of necroptosis assessed by caspase inhibition with Z-VAD is shown. In all other instances where the inhibitory effect of Cal/09 or the potentiating effect of NC/99 on cell death and T cell proliferation is evaluated (Figs. 4-6 and elsewhere), treatments with viruses and/or poly I:C are in the absence of Z-VAD. Without caspase inhibition in these experiments, how do the authors know that cells are undergoing immunogenic / necroptotic death as opposed to non-immunogenic apoptotic death? Granted that seasonal strains and poly I:C can induce necroptosis (3A, 4B), but how does one know that seasonal strains are indeed inducing necroptosis and pandemic strains are inhibiting necroptosis (and not another form of death) in these experiments? This is one of the primary claims of the study and there are no solid data in the paper to support this main conclusion. Fig. 3C implies that NC/99 is more efficient than Cal/09 at inducing both MLKL phosphorylation (necroptosis) and caspase 8 cleavage (apoptosis), and Fig. 3A shows that MLKL inhibitors can inhibit apoptosis. Taken together the data are thoroughly confusing. It is obscure how the authors can draw any sound conclusions about modulation of necroptosis by seasonal and pandemic IAV from these data.>>

Authors' Response: "Without caspase inhibition in these experiments, how do the authors know that cells are undergoing immunogenic / necroptotic death as opposed to non-immunogenic apoptotic death?" While seasonal IAV activates both MLKL and caspase 8 (Fig. 3c,d), we demonstrate that the cellular response is immunogenic cell death. Staurosporine-induced apoptosis in DC did not result in a significant activation of T cell proliferation by naïve DC (Supplementary Fig. 6). Thus, caspase-mediated apoptotic cell death is not immunogenic. In contrast, cell death induced by seasonal IAV infection (or poly I:C) is immunogenic and thus is consistent with necroptosis.

"how does one know that seasonal strains are indeed inducing necroptosis and pandemic strains are inhibiting necroptosis (and not another form of death) in these experiments?" The seasonal strain induces MLKL phosphorylation and, as noted above, an immunogenic cell death, which is in line with necroptosis. Furthermore, we show in new data presented in Supplementary Fig. 5c that Z-VAD treatment has no effect on either poly I:C-induced cell death (indicating it is not caspase-dependent) or on the inhibitory effect of Cal/09 on poly I:C-induced cell death. Overall, these data support the notion that the pandemic strain inhibits seasonal IAV-induced necroptosis. Our chimeric virus experiments (e.g. Fig. 6c) and new CRISPR ablation experiments (Fig. 3b, Supplementary Fig. 3) further support this mechanistic interpretation.

Overall, the data provide a coherent picture where RIPK3-MLKL-mediated immunogenic cell death is induced by seasonal virus infection and is inhibited by the pandemic HA segment. Genetic ablation of both RIPK3 and MLKL results in inhibition of RIPK3-MLKL-mediated immunogenic cell death.

<< 5. Figure S4: Conclusions are at odds with the data. Text on Pg. 6, line 174 reads: "Comparison of the effects of direct cell contact of naïve DC and infected DC (mixed DC co-culture) to that of only factors secreted

by infected DC (transwell DC culture) shows that after seasonal IAV infection, direct contact leads to an increase in CD86, CCR7 and HLA-ABC levels in the naïve DC (Supplementary Fig. 4). In contrast, the effects of exposure to DC infected with the pandemic Cal/09 infection, which blocks DC necroptosis, are restricted to only an increase in CCR7 levels in naïve DC; this increase was not enhanced by direct contact. No increases in the other maturation markers assayed in naïve DC were seen after direct or trans well exposure to pandemic virus-infected DC". Upregulation of markers except CCR7 in NC/99 mixed DC co-cultures is at best modest. Moreover, the flow plots as presented, show that direct contact of pandemic Cal/09 infected DCs with naïve DCs upregulates not only CCR7 but also CD86 and HLA-ABC in naïve DCs. Therefore, the authors conclusions are not supported by their data.

Authors' Response: In order to observe a better upregulation of maturation markers, we increased the co-culture incubation period from 8 hours to 18 hours for examining the effect of direct contact with dying cells vs. the effect of paracrine signaling via a transwell system (New Fig. 5). In these new experiments, we demonstrate a significant increase in HLA-DR, HLA-ABC, CD86, CD40 and CCR7 in DC in direct contact with NC/99-infected cells as compared to DC in contact with Cal/09-infected cells. By contrast, no significant increase in those maturation markers was observed when DC were separated from virus-infected cells by a transwell system. Therefore, our data demonstrate that direct contact with seasonal IAV-infected DC is necessary for an increase in maturation markers in uninfected DC.

<< 6. Pg 21, Figure 4g legend. last line reads: "g shows representative imaging flow cytometry plots". Is this correct or are the CFSE dilution plots regular flow plots?

Authors' Response: T cell proliferation was assayed by imaging flow cytometry, as it generates higher resolution data than standard flow cytometry. This is now indicated in the legend of Fig. 4.

<< 7. Figure 6: The conclusion that pandemic IAV causes reduced T cell proliferation is not supported by the data. In Fig. 6E, although more number of T cells incubated with naïve DCs crossprimed by NC/99 infected DCs enter division compared to T cells incubated with naïve DCs crossprimed by Cal/09 infected DCs, the latter divide more. The data as presented show that T cells incubated with DCs crossprimed by pandemic Cal/09 infected DCs that do enter division are more efficient proliferators compared to their NC/99 counterparts.>>

Authors' Response: We think the reviewer's comments are actually directed at Fig. 6b (now Fig. 7b). However, we disagree with the reviewer's interpretation of these data. Wherever the curves overlay, the green Cal/09 curve is plotted last, giving it more prominence. The T cell proliferation patterns induced by mock-infected and Cal/09-infected DC are indistinguishable graphically (Fig. 7b) and statistically (Fig. 7a). Thus, the data support the absence of an effect of Cal/09-infected DC on T cell proliferation. Furthermore, the major finding is that the proportion of unproliferated cells is clearly reduced by NC/99- and chimeric Cal/09-NC HA/NA IAV infection as compared to mock infection (or Cal/09 infection). Finally, the entire left curve of T cell proliferation obtained after mock or Cal/09 infection is contained within the larger proliferation curves obtained after NC/99 or Cal/09-NC HA/NA infection. If Cal/09 (or mock) infection caused more effective T cell proliferation, the curves would extend to the left beyond that of NC/99. However, we see more NC/99-induced T cell proliferation with all the curves sharing the left-hand border. This interpretation is also supported by the new maturation marker expression analysis (Fig. 5).

<< 8. *Figure 6C-D: It is unclear what the authors are trying to accomplish here. Where does the inhibitory effect of Cal/09 on T cell proliferation come from? The claim that DC markers are affected upon direct contact with DCs infected with NC/99 vs Cal/09 (Fig. S4 discussed previously) is unconvincing. The ability of DCs to make cytokines, which may provide insight into the mechanism of Cal/09 inhibition, is not evaluated.>>*

Authors' Response: We now show that direct contact of uninfected DC with DC infected with cell death inducing NC/99 express significantly higher levels of HLA-DR, HLA-ABC, CD86, CD40, and CCR7 than DC in contact with Cal/09-infected DC (Fig. 5). Additionally, we now include cytokine production assays (Fig. 2a): with the exception of IFN β and IP-10, cytokine expression levels are quite similar between seasonal and pandemic strains. Therefore, we conclude that the direct exposure to DAMPs and PAMPs from seasonal virus-infected DC induces maturation marker expression in uninfected DC. Cal/09, as described above, suppresses this immunogenic cell death and thereby represses T cell activation (Fig. 7c).

<< 9. *Figure 6E: These data remain unconvincing. I agree that they may reach statistical significance, but the heterogeneity in the post Bri/07 condition where CD8 counts in 3 out of 9 individuals completely crash make it very difficult to believe that the data are biologically significant. If the authors' in vitro data is to any degree reflective of the in vivo situation in humans, one would expect to see greater T cell proliferation aka greater numbers of circulating T cells in the post-BRI condition, which is not the case. This doesn't mean that the in vitro data is not useful, just that the human data is not at a point to draw any conclusions that support the authors' other data.>>*

Authors' Response: The reviewer raises two important concerns about: 1) overall changes that differ from what he/she would expect, 2) the effect of individual outliers on the validity of the interpretation of the Brisbane data.

Regarding the first point: our analysis shows that the relative circulating levels of T cells after infection with pandemic Cal/09 are significantly lower than before infection. This is not the case following Bri/07 infection. It is well established that during an infection, T cells leave the bloodstream and accumulate at the site of infection. Thus, if there was insufficient replenishment of this efflux of T cells from the blood (e.g. replenishment due to T cell proliferation and release into the blood), the circulating T cell levels would decrease. This appears to be what we see after Cal/09 infection. After virus infection, there is a) an increased recruitment of T cells to the site of infection (which by itself would reduce circulating T cell levels) and an increased production of T cells (which by itself would increase circulating T cell levels). If the rate of T cell production, as we presume, is lower after pandemic Cal/09 infection, then we expect to see a larger difference in T cell levels after infection for the pandemic virus than for the seasonal virus. We cannot predict whether T cell levels should go up or down after infection with each virus, as it depends on the overall changes in T cell migration from the blood and delivery into the blood following infection. We can only predict that the difference between pre- and post-infection T cell levels should be smaller after seasonal virus infection than after pandemic virus infection. Indeed, we find that while this difference is positive after pandemic infection, it is close to zero after seasonal infection. These results are equally consistent with the underlying model and hypothesis as with the specific pattern anticipated by the reviewer. Nevertheless, they do not determine unequivocally the mechanism underlying this difference. Furthermore, we could not identify any other appropriate seasonal IAV infection dataset in humans to independently confirm these findings. We have revised the text to emphasize that further study is required to confirm these findings and to determine whether these results on circulating T cells truly result from a differential T cell production rate after virus infection.

Regarding the second point: we apologize that the data were presented in a manner that appeared much less convincing than it is. The analyzed data comprise pooled data with three measurements per subject around peak symptoms (45.5, 53, and 60 hours after infection). The three “crash” points after Brisbane infection represent three measurements from the same individual, not three measurements from the nine individuals. This one individual is a clinical outlier as well, having the absolute highest symptom levels of all infected subjects. The results are unaffected by excluding this single subject (as well as by analyzing the data separately at each time point), but for transparency they are still included in the figure. We have redrawn the figure to show all data points and revised the legend to make the source of the plotted data clearer. While, as noted above, these results cannot be replicated in another dataset (as it doesn’t exist), when properly presented and discussed they are rather impressive and should remain in the study.

<< 10. *Supplementary fig. 1 is not cited in the text, only in the methods section.*>>

Authors’ Response:.

We now cite Supplementary Fig. 1 in the text.

Authors’ Response:.

<< 12. *Pg. 23, line 786: Supplementary Figure 6 should be Supplementary Figure 5. Pg. 24, line 794: Supplementary Figure 5 should be Supplementary Figure 6. Line 796-797: Delete “c Comparison of T cell proliferation by necroptotic and apoptotic cells.”>>*

Authors’ Response:.

These have been corrected.

Reviewers' comments:

Reviewer #2 (Remarks to the Author):

All previous concerns have been addressed.

Reviewer #3 (Remarks to the Author):

Hartmann et al provide additional data and clarifications to support some of the claims made in their previous report. While some of these strengthen the manuscript for instance CRISPR mediated genetic ablation of cell death genes, some questions remain (below) and DC maturation data as the underlying mechanism for differences between Cal09 and NC99 mediated T cell proliferation remain unconvincing. My outstanding concerns are:

1. Page 5: "To evaluate role of extrinsic pathways.....measured expression of cytokines involved in cell death induction including those known to be involved in cell death induction" Can the authors provide a citation for this statement?

2. p5: "No seasonal virus-specific increase in cytokines was observed (Fig. 2a)" However Fig. 2A shows increased expression of IFN- β , IP-10 and TNF- α with seasonal viruses. The authors mention this in the response to reviewers but it should be acknowledged earlier in the text, and a stronger discussion of these differences is warranted.

3. Values for some cytokines in Fig. 2 are at levels much below their detection limit by ELISA (e.g. 1.5-2 pg for IL-1 α and IL-1 β). It is unclear from the methods section or elsewhere how the authors achieved such detection.

4. Fig 3b, related methods and discussion of the results: Guide RNA sequences for genetically ablated genes + non-targeting gRNA sequences, and information on how many gRNAs were tested per gene should be provided. It is unclear if DCs were electroporated with non-targeting gRNA as the control. This should be clarified and is an essential control given the recently reported off target effects of CRISPR.

5. The authors discuss that the inhibitor NSA has off target effects and does not phenocopy the results obtained with genetic ablation of MLKL (Fig 3a and 3b). They suggest that NSA interferes with RIPK3 mediated caspase activation (as discussed on p6). However, they go on to use NSA in Fig. 4b and draw conclusions about MLKL involvement. To draw this conclusion from Fig. 4b the lane treated with NSA should also have Z-VAD, which doesn't seem to be the case.

6. Fig 4a and 4b: x-axis units for poly I:C concentration need to be corrected.

7. Data on DC maturation being a possible mechanism for differences in T cell proliferation between NC99 and Cal09 remain unconvincing. On Pg 8-9: The authors state that "merely direct contact leads to a significant increase in maturation markers in naïve DC. No

significant increase in maturation markers assayed in naive DC was observed after either direct contact or transwell exposure to pandemic Cal09 virus infected DC” But the representative histograms in Fig. 5a show what appears to be a significant upregulation of all markers with Cal09 compared to the mock condition. If upregulation of DC maturation markers was the underlying mechanism for T cell activation one would expect to see some T cell proliferation by Ca09 infected DC in for example supplementary Fig. 6 or Fig. 7a-b. On the other hand, the increase in marker expression on DC in direct contact with NC99 vs Cal09 infected DC seems to be quite modest (although the histograms aren’t overlaid as such, so it is hard to compare with the quantification in Fig. 5b). However according to the authors this difference can account for increased T cell proliferation. Overall, it isn’t convincing that differences in DC marker expression could be the underlying mechanism for increased T cell proliferation with NC99 compared to Cal09 infected DC. At the least the authors need to more strongly discuss how/why they think this could be the case.

8. Discussion p11: “This maturation effect requires direct contact between the infected and uninfected DC and is not seen when cells are separated by a cytokine permeable transwell membrane” Again, this statement is at odds with data in Fig.5 which shows increased maturation markers upon transwell coculture. In general, the conclusion that transwell culture has no effect is not supported by the data. The authors should be mindful of this, re-state and re-discuss what their data actually shows.

Response to Reviewers

Authors' Response Introduction:

We have provided supplementary experimental data and made revisions to Supplementary Fig. 5, Fig. 4, and to the main text to address each of the reviewer's comments. All changes to the main text are highlighted in red font. The reviewer's comments are quoted below in italics, while our responses are in plain text and follow each comment.

Reviewer #2:

All previous concerns have been addressed.

Reviewer #3:

Hartmann et al provide additional data and clarifications to support some of the claims made in their previous report. While some of these strengthen the manuscript for instance CRISPR mediated genetic ablation of cell death genes, some questions remain (below) and DC maturation data as the underlying mechanism for differences between Cal09 and NC99 mediated T cell proliferation remain unconvincing. My outstanding concerns are:

1. Page 5: "To evaluate role of extrinsic pathways.....measured expression of cytokines involved in cell death induction including those known to be involved in cell death induction" Can the authors provide a citation for this statement?

Authors' Response:

We have added several references in the main text supporting the involvement of the analyzed cytokines in cell death induction:

- Elmore 2007 PMID 17562483: TNF α is a ligand involved in the extrinsic apoptotic pathway.
- Hoorens 2001 PMID 11246874; Steer 2006 PMID 16354107: IL-1 induces cell death in beta-cells.
- von Karstedt 2017 PMID 28536452: TRAIL induces apoptosis.
- Chawla-Sarkar 2003 PMID 12766484: IFN α , IFN β , IFN γ induce apoptosis; TRAIL has an apoptotic function.
- Liu 2011 PMID 21802343: CXCL10 (aka IP-10) is involved in apoptosis induction.
- Mellado 2001 PMID 11369232: CCL3 (aka MIP-1a) triggers tumor-infiltrating lymphocyte (TIL) cell death.

2. p5: "No seasonal virus-specific increase in cytokines was observed (Fig. 2a)" However Fig. 2A shows increased expression of IFN-b, IP-10 and TNF-a with seasonal viruses. The authors mention this in the response to reviewers but it should be acknowledged earlier in the text, and a stronger discussion of these differences is warranted.

Authors' Response:

TNF α and IP-10 were higher following the pandemic virus infection, so these changes do not support the formulation that cell death after seasonal virus infection is mediated by paracrine signaling. We have revised the text in response to the reviewer's comments to try to make these points clear (main text on p.5):

“Overall, [cytokine] levels were comparable between seasonal and pandemic IAV infection, with the exception of IFN β . While IFN β was markedly induced by seasonal NC/99, it was weakly induced by pandemic Cal/09 and by the other seasonal strain Tx/91 (**Fig. 2a**). Thus, the induction of IFN β is not associated with IAV-induced dendritic cell death. Moreover, the absolute IFN β levels observed were low compared to the levels detected following infection with Newcastle disease virus³¹, which is a model system for immune activation³². The levels of TNF α and IP-10 tended to be higher following pandemic IAV infection than following cell death-inducing seasonal IAV infection. As the increases in both cytokines were seen with the pandemic virus, these results help to exclude the possibility that paracrine signaling contributes to the cell death induced by seasonal virus infection. As a further exclusion of this possibility, we show that the presence of a TNF α antagonist or of neutralizing antibodies against TRAIL or FASL did not reduce NC/99-induced cell death (**Supplementary Fig. 2**), thus supporting the conclusion that external signaling does not play a significant role in cell death induction by seasonal IAV infection.”

3. Values for some cytokines in Fig. 2 are at levels much below their detection limit by ELISA (e.g. 1.5-2 pg for IL-1a and IL-1b). It is unclear from the methods section or elsewhere how the authors achieved such detection.

Authors' Response:

The reviewer is correct that IL-1a and IL-1b cytokine levels are below the detection threshold for multiplex ELISA. As the point of this assay is to test for cytokines that might be having an effect on cell death, the data obtained, including levels below reliable assay measurement, are presented as they support the view that cytokines are not playing any role. To point this out, we have added the following sentence on p.5:

“Cytokine levels were generally low and in several cases below the limit of detection for the assay.”

Additionally, we have revised the legend of Fig. 2a to include the statement that:

”The levels of IL-1a, IL-1b, TRAIL, and IL-6 are near or below the limits of detection for this assay.”

Finally, we have added the following sub-section in the Methods:

“Cytokine measurements

The levels of all cytokines, except IFN β , secreted upon DC infection with either pandemic or seasonal IAV strains were measured using a multiplex ELISA kit, following the manufacturer's

recommendations (EMD Millipore). IFN β was measured using a conventional ELISA kit (PBL Interferon Source).”

4. Fig 3b, related methods and discussion of the results: Guide RNA sequences for genetically ablated genes + non-targeting gRNA sequences, and information on how many gRNAs were tested per gene should be provided. It is unclear if DCs were electroporated with non-targeting gRNA as the control. This should be clarified and is an essential control given the recently reported off target effects of CRISPR.

Authors' Response:

Two different guide RNAs were tested for each gene and the guide RNA giving the best ablation was selected for additional study. We provide in the Methods the primer sequences for all guide RNAs that were used. As a negative control, DC were electroporated with Cas9 Nuclease in the absence of guide RNAs. This is now indicated in the figure legend.

Because off-target effects are sequence-specific, non-targeting guide RNAs do not provide a control for off-target binding of the tested guide RNAs. In our hands, DC that were electroporated with Cas9 ribonucleoprotein complexes and then virus-infected were gated based on lower protein expression of the targeted gene(s) prior to assaying cell death (Supplementary Fig. 3). Compared to a bulk analysis of all electroporated cells, this gating strategy reduces the capture of potential off-target effects. Furthermore, it is noteworthy that the newer ribonucleoprotein approach that we utilized was developed to eliminate off-target effects and has been shown to be highly specific. In contrast to the previous vector-based CRISPR approaches, the ribonucleoprotein method is short-lived in the cell and dramatically reduces off-target effects (Liang 2015 PMID 26003884). In addition, the studies with structurally distinct chemical inhibitors serve as additional independent evidence that support the major conclusions from these mechanistic studies.

5. The authors discuss that the inhibitor NSA has off target effects and does not phenocopy the results obtained with genetic ablation of MLKL (Fig 3a and 3b). They suggest that NSA interferes with RIPK3 mediated caspase activation (as discussed on p6). However, they go on to use NSA in Fig. 4b and draw conclusions about MLKL involvement. To draw this conclusion from Fig. 4b the lane treated with NSA should also have Z-VAD, which doesn't seem to be the case.

Authors' Response:

As requested, we now provide experimental data in panel (a) of Supplementary Fig. 5, showing the effect of Nec-1, NSA, Z-VAD, or the combination of Z-VAD with either Nec-1 or NSA on poly I:C-induced cell death. The results show that poly I:C-induced cell death is significantly reduced by the MLKL inhibitor NSA both in the presence and absence of pan-caspase inhibitor Z-VAD. Conversely, RIPK-1 inhibitor Nec-1 has no significant effect on poly I:C-induced cell death, whether Z-VAD is concomitantly present or not. These results complement the data depicted in Fig. 4b and are consistent with the induction of an MLKL-dependent, RIPK1-

independent cell death pathway by seasonal IAV infection (Fig. 3a). Note that we have also modified the order of panels (a-c) in Supplementary Fig. 5 to reflect the order in which they are cited in the main text.

6. *Fig 4a and 4b: x-axis units for poly I:C concentration need to be corrected.*

Authors' Response:

We now provide a corrected version of Fig. 4.

7. *Data on DC maturation being a possible mechanism for differences in T cell proliferation between NC99 and Cal09 remain unconvincing. On Pg 8-9: The authors state that “merely direct contact leads to a significant increase in maturation markers in naïve DC. No significant increase in maturation markers assayed in naïve DC was observed after either direct contact or transwell exposure to pandemic Cal09 virus infected DC” But the representative histograms in Fig. 5a show what appears to be a significant upregulation of all markers with Cal09 compared to the mock condition. If upregulation of DC maturation markers was the underlying mechanism for T cell activation one would expect to see some T cell proliferation by Ca09 infected DC in for example supplementary Fig. 6 or Fig. 7a-b. On the other hand, the increase in marker expression on DC in direct contact with NC99 vs Cal09 infected DC seems to be quite modest (although the histograms aren't overlaid as such, so it is hard to compare with the quantification in Fig. 5b). However according to the authors this difference can account for increased T cell proliferation. Overall, it isn't convincing that differences in DC marker expression could be the underlying mechanism for increased T cell proliferation with NC99 compared to Cal09 infected DC. At the least the authors need to more strongly discuss how/why they think this could be the case.*

Authors' Response:

We appreciate the reviewer's careful reading and apologize for any lack of clarity. We address the reviewer's concerns point by point below:

i) *The authors state that “merely direct contact leads to a significant increase in maturation markers in naïve DC”.*

We have corrected/rephrased this sentence in the text to address the reviewer's concern:

“While we observed a dramatic upregulation of DC maturation markers when NC/99-infected DC were in direct cell-to-cell contact with uninfected DC, this upregulation was much more modest when NC/99-infected DC were separated from uninfected DC by a transwell membrane.”

ii) *“No significant increase in maturation markers assayed in naïve DC was observed after either direct contact or transwell exposure to pandemic Cal09 virus infected DC”*

We appreciate the reviewer noting that this is phrased imprecisely. What this experiment demonstrates is that the difference between contact effects are dramatically greater following the NC/99 seasonal infection in comparison with the pandemic Cal/09 infection. We have eliminated this sentence and replaced it with:

“There was little difference in the effects of pandemic Cal/09 vs. seasonal NC/99 infection on maturation marker expression in uninfected DC separated from the infected cells by a transwell membrane. In contrast, direct contact with NC/99-infected cells caused a significant increase in nearly all maturation markers studied relative to direct contact with Cal/09-infected cells.”

iii) *If upregulation of DC maturation markers was the underlying mechanism for T cell activation one would expect to see some T cell proliferation by Ca09 infected DC in for example supplementary Fig. 6 or Fig. 7a-b.*

We agree with the reviewer that there is a slight increase of activation markers with exposure to Cal/09 infected cells and that this is insufficient to cause T cell activation. This is not inconsistent with the significantly greater maturation marker induction with exposure to NC/99-infected cells contributing to T cell activation by those cells. Many immune system processes have a threshold effect requiring multiple activation of simultaneous gating signals (e.g. signal I, signal II, etc.), and the exposure to Cal/09-infected cells does not induce a level of DC maturation sufficient to contribute to measureable T cell proliferation. As noted in our response below, we are not claiming that this is the only mechanism involved, but the results are certainly consistent with this being the major factor.

iv) *On the other hand, the increase in marker expression on DC in direct contact with NC99 vs Cal09 infected DC seems to be quite modest (although the histograms aren't overlaid as such, so it is hard to compare with the quantification in Fig. 5b). However according to the authors this difference can account for increased T cell proliferation.*

In direct contact co-cultures, the difference in the increased expression of DC maturation markers between NC/99 and Cal/09 infection is statistically significant. However, we agree with the reviewer that additional mechanisms besides increased DC maturation may contribute to the rise in T cell proliferation in the NC/99 infection scenario. For instance, NC/99-infected DC could directly activate T cells (direct priming) more effectively than Cal/09-infected DC. *In vivo*, necroptotic DC may also “alert” other local antigen-presenting cells (e.g. macrophages) via the release of DAMPs, making them stimulatory to the adaptive immune system. Based on our experimental data, direct cell-to-cell interaction between NC/99-induced dying DC and uninfected DC plays a significant role in activation of the uninfected DC. Although we cannot rule out the possible contribution of other mechanisms, investigating those is beyond the scope of the present study.

We have edited the concluding sentence of the top paragraph on p.9 in the Results:

“Overall, these results suggest that direct cell-to-cell interactions between NC/99-induced dying DC and uninfected DC significantly contribute to activation of the uninfected DC, thus associated with an increase in MHC, costimulatory molecules, and CCR7 homing signaling.”

We have also edited the Discussion on p.13 to address the reviewer’s concerns:

“Using co-culture and transwell experiments, we find that necroptotic seasonal IAV-infected DC are a potent stimulus for initiating DC maturation (see **Fig. 5**). This maturation effect is heightened by direct contact between the infected and uninfected DC, while it is much more moderate when the cells are separated by a cytokine-permeable transwell membrane. Nonetheless, additional mechanisms may contribute to the increase in T cell proliferation following seasonal IAV infection. For example, it is possible that NC/99-infected DC may directly activate T cells (direct

priming) more effectively than Cal/09-infected DC. *In vivo*, necroptotic DC may alert other local antigen-presenting cells (e.g. macrophages) via the release of DAMPs, rendering those cells stimulatory to the adaptive immune system.”

8. *Discussion p11: “This maturation effect requires direct contact between the infected and uninfected DC and is not seen when cells are separated by a cytokine permeable transwell membrane” Again, this statement is at odds with data in Fig.5 which shows increased maturation markers upon transwell coculture. In general, the conclusion that transwell culture has no effect is not supported by the data. The authors should be mindful of this, re-state and re-discuss what their data actually shows.*

Authors’ Response:

Please see our response to reviewer’s comment #7 ii).

REVIEWERS' COMMENTS:

Reviewer #3 (Remarks to the Author):

All comments have been addressed.